# EXTRA-NEWTON: A First Approach to Noise-Adaptive Accelerated Second-Order Methods

**Kimon Antonakopoulos** *
EPFL (LIONS)
kimon.antonakopoulos@epfl.ch

**Ali Kavis**\*
EPFL (LIONS)
ali.kavis@epfl.ch

**Volkan Cevher**
EPFL (LIONS)
volkan.cevher@epfl.ch

## Abstract

This work proposes a universal and adaptive second-order method for minimizing second-order smooth, convex functions. Our algorithm achieves $O(\sigma/\sqrt{T})$ convergence when the oracle feedback is stochastic with variance $\sigma^2$, and improves its convergence to $O(1/T^3)$ with deterministic oracles, where $T$ is the number of iterations. Our method also interpolates these rates without knowing the nature of the oracle apriori, which is enabled by a parameter-free adaptive step-size that is oblivious to the knowledge of smoothness modulus, variance bounds and the diameter of the constrained set. To our knowledge, this is the first universal algorithm with such global guarantees within the second-order optimization literature.

## 1   Introduction

Over the last few decades, first-order (convex) minimization methods have gained popularity for modern machine learning and optimization problems due to their efficient per-iteration cost and *global convergence* properties. The literature on first-order methods is rather dense and extensive with a concrete, thorough understanding of the optimal *global* convergence behavior. Focusing on the more relevant settings of smooth, convex minimization, the lower bounds have been well-established; $O(\sigma/\sqrt{T})$ when the gradient feedback is noisy with variance $\sigma^2$, and $O(1/T^2)$ under deterministic first-order oracles [51, 57]. Under slight variations of the aforementioned problem setting, there exists an extensive amount of work that enjoys the latter, "accelerated" rate [55, 56, 52, 64, 68, 39, 2, 41, 66, 19, 18, 35, 31, 6, 44].

On the contrary to its first-order analogue, the literature on *global convergence* of second-order, smooth methods is notably sparse with many open questions standing even in the simplest problem formulations. Following the pioneering works of Bennett [11], Kantorovich [33], Newton's method and its variations [40, 46] are considered as the staple of second-order methods in optimization. Although its powerful local convergence behavior has been repeatedly demonstrated [17, 38], studies on its global behavior are relatively limited. Prior attempts at tackling global convergence mostly make additional structural assumptions on the objective function [60, 47, 38] or assume extra regularity conditions on the Hessian [34] beyond the simplest smooth and convex setting. Over the last decade, we have witnessed important progress towards a more complete theory of globally-convergent second-order methods (more on this shortly), and yet there remains many important questions unanswered, which we will delve into in this paper.

To motivate the perspective in our technical endeavour, we take a small detour to introduce the idea of *universality*, which we particularly characterize as *adaptation to the level of noise in oracle feedback*. Enabled by the recent advances in online optimization, universal first-order algorithms essentially attain the $O(\sigma/\sqrt{T} + 1/T^2)$ convergence for convex minimization problems, interpolating between stochastic and deterministic rates. There exist a plethora of algorithms that enjoy this rate under

---

*Alphabetical order, equal contribution

36th Conference on Neural Information Processing Systems (NeurIPS 2022).

different sets of assumptions for both minimization scenarios (for convex and non-convex settings, we refer the reader to [39, 35, 22, 31, 6] and [67, 42, 36, 45], respectively), and the more general framework of variational inequalities [8, 5, 65, 25, 3, 26, 4]. However, we observe that such universal results do not exist in second-order literature, hence, it is only natural to ask,

*Can we design a simple second-order method that will achieve*
*accelerated universal rates beyond $O(\sigma/\sqrt{T} + 1/T^2)$?*

More recently, global sub-linear convergence rates for second-order methods have been characterized by [58] for second-order smooth and convex setting. Essentially, the so-called Cubic Regularized Newton's Method combines the quadratic Taylor approximation in the typical Newton update with a cubic regularization term. At the expense of solving a cubic problem, this method achieves $O(1/T^2)$ convergence rate. Shortly after, Nesterov [54] proposes an accelerated version of the cubic regularization idea with $O(1/T^3)$ value convergence, pioneering a new direction of research in the study of globally-convergent second-order methods [48]. This idea has been studied further for different settings in convex optimization [28, 29] with the same accelerated $O(1/T^3)$ rate and extended to non-convex realm [14, 15], obtaining the analogous rates of $O(1/T^{2/3})$ and $O(1/T^{1/3})$ for finding first-order and second-order stationary points, respectively, leading the way for further investigations [10, 21, 16].

Notice that accelerated cubic regularization is *sub*-optimal such that recent studies prove a respective lower-bound for second-order smooth, convex problems as $O(1/T^{7/2})$ [1, 7]. The first line of research that shrinks the gap between the upper and lower bounds for achieving an *almost*-optimal (more on this shortly) convergence [59] is the so-called "bisection-type" methods. Pioneered by Monteiro and Svaiter [49], these class of algorithms propose a conceptual method where the step-size of the algorithm *implicitly* depends on the next iterate. To resolve, the authors propose a bisection procedure that simultaneously finds a step-size/next iterate pair that satisfies the conditions of the iterative update, which enables the convergence rate of $O(1/T^{7/2})$, modulo the complexity of bisection procedure. This idea was very recently generalized for higher-order tensor methods [23]. Not so surprisingly, the same construction finds application in variational inequality (VI) and min-max optimization literature [12, 30]. Very recently and concurrently to our work, [13] propose the first bisection free acceleration for second-order methods, that achieves the optimal $O(1/T^{7/2})$. The authors define an *explicit*, deterministic procedure called MS oracle and compute the step-size using a standard line-search procedure enabling them to achieve optimal rates while adaptively computing the step-size without needing to know the smoothness constant.

Although there are promising results with an increasing interest into second-order –and also higher-order– methods, we identify three main shortcomings in the literature, which we will systematically address in the sequel. First, bisection-type methods achieve the optimal convergence rate however, the search procedure is computationally very prohibitive [59, 43] and the resulting algorithms are complicated with many interconnected components. On the other hand, cubic regularization-based ideas propose a simple construction that achieves acceleration beyond $O(1/T^2)$ however, similar to previous methods, they either require the knowledge of smoothness constant or need to execute a standard line-search procedure to estimate it locally. A common drawback for both approaches is that the algorithmic constructions are designed for handling *only* deterministic oracles and it is an open question whether such frameworks could immediately accommodate stochastic first and second-order information.

**Our contributions:** To address the aforementioned issues, we developed the first universal and adaptive second-order algorithm, EXTRA-NEWTON, for convex minimization. We summarize our contributions as follows:

1. We prove EXTRA-NEWTON achieves the global convergence rate of $O(\frac{\sigma_g}{\sqrt{T}} + \frac{\sigma_H}{T^{3/2}} + \frac{LD^3}{T^3})$ that adapts simultaneously to the variance in the gradient oracle ($\sigma_g$) and Hessian oracle ($\sigma_H$) achieving the first universal convergence result in the literature.

2. Our method is completely oblivious to any problem-dependent parameters including smoothness modulus, variance bounds on stochastic oracles, diameter of the constraint set and any possible bounds on the gradient and Hessian.

**Table 1:** A survey on first and second-order algorithms with key properties

| | AGD [55] | UniXGrad [35] | Reg. Newton [48] | Accel. Cubic Reg. [54] | ANPE[2] [49] | OptMS [13] | Extra Newton [ours] |
|---|---|---|---|---|---|---|---|
| *Rate* | $\frac{1}{T^2}$ | $\frac{\sigma_g}{\sqrt{T}} + \frac{1}{T^2}$ | $\frac{1}{T^2}$ | $\frac{1}{T^3}$ | $\frac{1}{T^{7/2}}$ | $\frac{1}{T^{7/2}}$ | $\frac{\sigma_g}{\sqrt{T}} + \frac{\sigma_H}{T^{3/2}} + \frac{1}{T^3}$ |
| *Bisection-free* | ✓ | ✓ | ✓ | ✓ | ✗ | ✓ | ✓ |
| *Adapts to L* | ✗ | ✓ | ✗ | Partial | ✗ | ✓ | ✓ |
| *Noise-adaptive* | ✗ | ✓ | ✗ | ✗ | ✗ | ✗ | ✓ |

3. We design the first adaptive step-size, in the sense of [20, 62], that successfully incorporates second-order information "on-the-fly". While doing so, we bypass any bisection or linesearch procedure, and propose a simple, intuitive algorithmic framework.

From a technical point of view, what will allow us to achieve these results is the combination of three principal ingredients: (*i*) proposing appropriate adjustments to Extra-Gradient [37] that was originally designed for solving variational inequalities and min/max problems; (*ii*) an "optimistic" weighted iterate averaging scheme accompanied by an appropriate gradient rescaling strategy in the spirit of [66, 19, 35] which allows us to obtain an accelerated rate of convergence by means of a generalized online-to-batch conversion (Theorem 3.3), and (*iii*) the glue that holds these elements together is an adaptive learning rate inspired by [62, 35, 4] which automatically rescales aggregated gradients and second order information. In what follows, we shall explicate these arguments.

## 2 Problem setup

Throughout the sequel, we will be focusing on solving (constrained) convex minimization problems of the general form:

$$\begin{aligned} \text{minimize} \quad & f(x) \\ \text{subject to} \quad & x \in \mathcal{X}. \end{aligned} \tag{Opt}$$

Formally, in the above $\mathcal{X}$ is a convex and compact subset of a $d$- dimensional normed space $\mathcal{V} \cong \mathbb{R}^d$ with diameter $D = \max_{x,y \in \mathcal{X}} \|x - y\|$, and $f : \mathcal{V} \to \mathbb{R} \cup \{+\infty\}$ is a proper, lower semi-continuous, convex function with $\text{dom} f = \{x \in \mathbb{R}^d : f(x) < +\infty\} \subset \mathcal{X}$. To that end, we make a set of blanket assumptions for (Opt). Following the vast literature of constrained convex minimization [53, 9], we consider "simple" constraint sets, i.e.,

**Assumption 2.1.** *The constraint set $\mathcal{X}$ of* (Opt) *possesses favorable geometry which facilitates a tractable projection operator.*

In order to avoid trivialities, we also assume that the said problem admits at least a solution, i.e.

**Assumption 2.2.** *The solution set $\mathcal{X}^* = \arg\min_{x \in \mathcal{X}} f(x)$ of* (Opt) *is non-empty.*

Furthermore, we assume that there exists a Lipschitz continuous selection $x \mapsto \nabla^2 f(x) \in \mathbb{R}^{d \times d}$, i.e.,

$$\|\nabla^2 f(x) - \nabla^2 f(x')\| \le L\|x - x'\| \quad \forall x, x' \in \mathcal{X} \tag{H-smooth}$$

and in addition it satisfies the second order approximation:

$$f(x) = f(x') + \langle \nabla f(x'), x - x' \rangle + \frac{1}{2}\langle \nabla^2 f(x')(x - x'), x - x' \rangle + O\left(\|x - x'\|^3\right) \tag{Taylor}$$

To that end, combining (H-smooth) and (Taylor) we readily get the following inequality:

$$\|\nabla f(x) - \nabla f(x') - \nabla^2 f(x')(x - x')\| \le \frac{L}{2}\|x - x'\|^2 \tag{1}$$

The above equivalences are well-established and hence we omit their proofs (we defer for a panoramic view to [69])

---

[2]Note that the bisection procedure is computationally prohibitive, we defer the reader to [59], p.304-305.

**Oracle feedback structure** From an algorithmic point of view, we aim to solve (Opt) by using methods that require access to a (stochastic) first and second order-oracle. Before we move forward with the methodology, we shall introduce the definitions and notations for this oracle model which we will use in algorithm definitions and technical discussions. Let $g(x, \xi)$ denote the stochastic gradient evaluated at $x$ with randomness defined by $\xi$ and $H(x, \xi)$ be the stochastic Hessian at $x$ with $\xi$ describing the randomness of the oracle, such that

$$
\begin{aligned}
\mathbb{E}\left[\, g(x, \xi) \mid x \,\right] = \nabla f(x), &\qquad \mathbb{E}\left[\, \|g(x, \xi) - \nabla f(x)\|^2 \mid x \,\right] \le \sigma_g^2 \\
\mathbb{E}\left[\, H(x, \xi) \mid x \,\right] = \nabla^2 f(x), &\qquad \mathbb{E}\left[\, \|H(x, \xi) - \nabla^2 f(x)\|^2 \mid x \,\right] \le \sigma_H^2
\end{aligned}
\tag{2}
$$

Due to space constraints, we will also define an operator that accommodates second-order information and its respective stochastic counterpart.

$$
\begin{aligned}
\mathbf{F}(x; x') &= \nabla f(x') + \frac{1}{2}\nabla^2 f(x')(x - x') \\
\tilde{\mathbf{F}}(x; x', \xi) &= g(x', \xi) + \frac{1}{2}H(x', \xi)(x - x')
\end{aligned}
\tag{3}
$$

where $\mathbf{F}$ is essentially the gradient (with respect to $x$) of the second-order Taylor polynomial. By definition, the operator $\mathbf{F}$ satisfies the second-order smoothness property in Eq. (1)

## 3 Method

In this section, we shall establish our universal second-order framework. Our presentation evolves around three key components: choosing the appropriate algorithmic template with the key motivations behind it, solving implementability issues that commonly arise in higher-order methods and finally designing a universal algorithm that can handle deterministic and noisy oracle feedback simultaneously without having prior knowledge. Our point of departure is the popular Extra-Gradient (EG) template; originally introduced by Korpelevich [37] and further developed in Nemirovski [50],

$$
\begin{aligned}
X_{t+\frac{1}{2}} &= \Pi_{\mathcal{X}}\left(X_t - \gamma_t \nabla f(x_t)\right) \\
X_{t+1} &= \Pi_{\mathcal{X}}\left(X_t - \gamma_t \nabla f(x_{t+1/2})\right),
\end{aligned}
\tag{EG}
$$

where $\Pi_{\mathcal{X}}(x) = \arg\min_{z \in \mathcal{X}} \|x - z\|^2$ is the standard Euclidean projection onto the set $\mathcal{X}$. In terms of output, the candidate solution returned by (EG) after $T$ iterations is the so-called "ergodic average"

$$
\bar{X}_T = \frac{\sum_{t=1}^{T} b_t X_{t+\frac{1}{2}}}{\sum_{t=1}^{T} b_t}
\tag{4}
$$

Then, taking $b_t = \gamma_t$ and assuming the method's step-size $\gamma_t$ is chosen appropriately, $\bar{X}_T$ enjoys the following universal guarantee [32, 61]:

$$
\mathbb{E}[f(\overline{X}_t) - f(x^*)] = \mathcal{O}\left(\frac{1}{T} + \frac{\sigma}{\sqrt{T}}\right)
\tag{5}
$$

where $\sigma$ signifies the effect of the noisy feedback. However, as it becomes apparent, the vanilla (EG) template is not capable of matching the iconic $1/T^2$ for the smooth deterministic case. It is well-established in the literature of smooth, convex minimization that iterate averaging (or momentum in the sense of Nesterov [55]) is essential for matching the $O(1/T^2)$ lower bounds. In fact, plain uniform averaging is not sufficient; one needs to introduce new iterates with *increasing* weights. Precisely, this is equivalent to computing an average by taking $b_t = O(t)$. However, we cannot fully characterize the acceleration machinery without what we like to call "gradient weighting". On top of (weighted) iterate averaging, gradients must be multiplied by the *same order of weights* to achieve acceleration, which is a recurring theme in the literature of accelerated and universal optimization [64, 68, 39, 2, 41, 66, 18, 35, 31].

Going back to discussion on (EG), Wang and Abernethy [66] and Kavis et al. [35] provide useful insights into acceleration within the context of (EG). Wang and Abernethy [66] identifies a 2-player game with a particular structure called FENCHELGAME framework, which essentially reduces to minimizing a smooth, convex function when the players cooperate. By introducing an "optimistic" weighted iterate averaging along with a complementary gradient weighting strategy, the framework

recovers different acceleration schemes of Nesterov [55, 56, 52]. On a related front, Diakonikolas and Orecchia [19] proposes the first acceleration of (EG) by appropriately integrating the optimistic averaging idea [66] into the (EG) template as follows:

$$\tilde{X}_t = \frac{b_t X_t + \sum_{s=1}^{t-1} b_s X_{s+\frac{1}{2}}}{\sum_{s=1}^{t} b_s}, \qquad \bar{X}_{t+\frac{1}{2}} = \frac{\sum_{s=1}^{t} b_s X_{s+\frac{1}{2}}}{\sum_{s=1}^{t} b_s} \qquad (6)$$

where $b_t = O(t)$ is the "iterate averaging" parameter. Later on, Kavis et al. [35] designs an adaptive, universal variant of accelerated Mirror-Prox following the same optimistic averaging idea as in Eq. (6). All in all, it is a recurring theme among accelerated algorithms to adopt weighted iterate averaging ($b_t = O(t)$) with proportionate gradient weighting, and not so surprisingly, prior work establishes clear connections between the degree of weighting and convergence rate. Cutkosky [18] designs a black-box reduction that accelerates a class of online algorithms and proves that the rate of convergence of the reduction is $O(1/\sum_{t=1}^{T} b_t)$ for $b_t \in [1, t]$. In retrospect, we aim at answering the following question;

*What algorithmic construction would enable acceleration beyond $O(1/T^2)$?*

## 3.1 Implicit algorithm

We give a first affirmative answer to the above question by presenting our implicit accelerated algorithm which is constructed upon (EG), and establish its convergence properties. Note that the implicitness of the scheme serves as a gentle introduction to the actual explicit second order acceleration, which shall follow. Formally, our scheme is given via the following recursion:

$$X_{t+\frac{1}{2}} = \Pi_{\mathcal{X}} \left( X_t - \gamma_t a_t \mathbf{F}(\bar{X}_{t+\frac{1}{2}}; \tilde{X}_t) \right)$$
$$= \arg\min_{x \in \mathcal{X}} a_t \langle \nabla f(\tilde{X}_t) + \frac{1}{2} \nabla^2 f(\tilde{X}_t)(\bar{X}_{t+\frac{1}{2}} - \tilde{X}_t), x - X_t \rangle + \frac{\|x - X_t\|^2}{2\gamma_t}$$

$$\text{(Implicit)}$$

$$X_{t+1} = \Pi_{\mathcal{X}} \left( X_t - \gamma_t a_t \nabla f(\bar{X}_{t+\frac{1}{2}}) \right)$$
$$= \arg\min_{x \in \mathcal{X}} a_t \langle \nabla f(\bar{X}_{t+\frac{1}{2}}), x - X_t \rangle + \frac{\|x - X_t\|^2}{2\gamma_t}$$

with $\Pi_{\mathcal{X}}(x)$ denoting the Euclidean projection of $x$ onto $\mathcal{X}$, average sequences $\tilde{X}_t$ and $\bar{X}_{t+\frac{1}{2}}$ defined as in (6) and the adaptive step-size $\gamma_t$ defined as (for some $\gamma, \beta_0 > 0$):

$$\gamma_t = \frac{\gamma}{\sqrt{\beta_0 + \sum_{s=1}^{t-1} a_s^2 \|\nabla f(\bar{X}_{s+\frac{1}{2}}) - \mathbf{F}(\bar{X}_{s+\frac{1}{2}}; \tilde{X}_s)\|^2}}. \qquad (7)$$

The implicit nature of (Implicit) originates from $X_{t+1/2}$ update (which we shall refer to as (corrected) extrapolation step at times) since $\bar{X}_{t+\frac{1}{2}}$ depends upon $X_{t+\frac{1}{2}}$ itself. However, this scheme exhibits several key differences from the vanilla (EG), which constitute the fundamental parts of our second-order acceleration machinery. In particular, we have:

(i) integration of second-order updates for sharper extrapolation steps - first step of acceleration.
(ii) interplay between averaging ($b_t$) and gradient weighting ($a_t$) which allows more aggressive averaging - second step of acceleration.
(iii) adaptive step-size in the sense of Rakhlin and Sridharan [62] - key to adaptivity and universality.

**Second-order updates:** First, we will consider the particular interpretation of (EG) as an approximation to the Proximal Point method [63] which serves as motivation for the accommodation of second-order information in our scheme.

$$X_{t+1} = X_t - \gamma_t \nabla f(X_{t+1}). \qquad \text{(PP)}$$

In particular, (EG) tries to approximate $X_{t+1}$ by generating the extrapolated point $X_{t+\frac{1}{2}}$, and make use of the gradient at $X_{t+\frac{1}{2}}$ to take a step from $X_t$ to $X_{t+1}$. Therefore, if the algorithm is able

to compute a sharper estimate in the extrapolation step, it should be able live up to the fame of (PP) and display faster convergence. To this end, we augment the extrapolation step by introducing second-order term. Essentially, our algorithm makes use of *second-order Taylor approximation*, as opposed to first-order expansion, only for the extrapolation step, trading-off sharper approximation with second-order information.

**Iterate averaging and gradient weighting:** Now, we turn our attention to the second component in our acceleration machinery; averaging and weighting. Recall that the acceleration framework of Cutkosky [18] guarantees a value convergence rate of $O(1/t^{p+1})$ when weighting factor satisfies $b_t = O(t^p)$ with $p \in [0, 1]$. We take this result one step beyond in two fronts; our algorithm exploits higher-order smoothness in order to extend this bound for $p \in [0, 2]$, implying the accelerated rate of $O(1/T^3)$. Second, we observe that previous work restricts the choice of gradient weights and averaging weights by taking $a_t \approx b_t$. We decouple those weights by allowing the sequences $a_t$ and $b_t$ to be *different*, which in turn equips us with more aggressive iterate averaging when necessary.

**Adaptive step-size:** As the final component, we study the adaptive step-size (7) from the parameter adaptation perspective (i.e., adaptation to the Lipschitz modulus) and expand on its universal properties in the next section. The vast literature on adaptive methods predominantly rely on constructions of AdaGrad-like decreasing step-size policies by accumulating the observed gradient norms in its denominator. The intuition behind this choice is that whenever the method approaches a solution, the vanishing gradients bring about stabilization, ensuring progress around the solution's neighborhood. However, this idea fails for (compactly) constrained problems; when the solution lies on the boundary. So inspired by [62], we design a constraint-aware step-size by accumulating $\|\nabla f(\bar{X}_{t+\frac{1}{2}}) - \mathbf{F}(\bar{X}_{t+\frac{1}{2}}; \tilde{X}_t)\|^2$ which converges to 0 as $\bar{X}_{t+\frac{1}{2}} - \tilde{X}_t \to 0$; which in turn implies convergence of the algorithm. To our knowledge, this is the first adaptive step-size that accommodates second order information.

Having established the core components of our design, we are in position to present the first accelerated convergence rate guarantee for (Implicit). Formally, this is given by the following.

**Theorem 3.1.** *Let* $\{X_{t+\frac{1}{2}}\}_{t=1}^T$ *be generated by* (Implicit) *run with the adaptive step-size policy* (7) *where* $a_t = t^2$, $b_t = t^p$ *with* $p \geq 2$. *Assume that* $f$ *satisfies* (H-smooth) *then, it is ensured that:*

$$f(\bar{X}_{T+\frac{1}{2}}) - f(x^*) \leq O\left(\frac{\max\left\{\sqrt{\beta_0}\frac{D^2}{\gamma}, L\frac{D^4+D\gamma^3}{\gamma}\right\}}{T^3}\right)$$

*When* $\gamma = D$, *we obtain the converge rate* $O\left(\frac{\max\left\{LD^3, \sqrt{\beta_0}D\right\}}{T^3}\right)$.

*Remark* 3.1. We emphasize that the above rate *does not* require any prior knowledge of problem paramaters such as $L$, $D$, time-horizon $T$ and any bounds on gradient/Hessian norms. In order to have better dependence on $D$ one could set $\gamma = D$, and our rate of $O(1/T^3)$ holds irrespective of $\gamma$.

### 3.2 Explicit algorithm

Despite the fact that (Implicit) improves upon the accelerated rate of $O(1/T^2)$, one may easily observe that it exhibits the following drawbacks:

1. (Implicit) is a conceptual algorithm and therefore, *not* implementable in practice.
2. A fortiori, it cannot provide rate interpolation guarantees as it does not have the machinery to simultaneously cope with deterministic and stochastic feedback.

As discussed earlier, a common strategy for overcoming this implicit construction is using a bisection/line-search procedure [30, 49, 12]. Depending on the context, this procedure serves two *distinct* purposes. Primarily, it tackles the implicit nature of the update rule by simultaneously finding a pair of $(\gamma_t, X_{t+\frac{1}{2}})$ and secondly, it enables adaptation to the second-order smoothness. However, one may identify major setbacks with these approaches; first, it is not clear how to handle stochastic oracles for executing the search procedure, so it is not capable of satisfying any universal guarantees. Moreover, it yields a rather complicated procedure as a byproduct that has many moving parts. To that end, we propose an alternative approach which not only yields a simple scheme, but

also provides a universal algorithm that is able to handle noisy feedback on-the-fly. Without further ado, we display our explicit algorithm, EXTRA-NEWTON, with appropriate modifications. Having defined our main scheme, Algorithm 1, we will provide a more detailed description of its components.

---

**Algorithm 1:** EXTRA-NEWTON

---

**Input**: $X_1 \in \mathcal{X}$, $a_t = t^2$ and $A_t = \sum_{s=1}^{t} a_s$, $b_t = t^p$ ($p \geq 2$) and $B_t = \sum_{s=1}^{t} b_s$, $\gamma > 0, \xi_t \sim$ i.i.d.

1: **for** $t = 1$ to $T$ **do**

2: $\qquad \gamma_t = \dfrac{\gamma}{\sqrt{\beta_0 + \sum_{s=1}^{t-1} a_s^2 \|g(\bar{X}_{s+\frac{1}{2}}, \xi_{s+\frac{1}{2}}) - \tilde{\mathbf{F}}(\bar{X}_{s+\frac{1}{2}}; \tilde{X}_s, \xi_s)\|^2}}$

3: $\qquad X_{t+\frac{1}{2}} = \arg\min_{x \in \mathcal{X}} \langle a_t g(\tilde{X}_t, \xi_t), x \rangle + \frac{a_t b_t}{2B_t} \langle H(\tilde{X}_t, \xi_t)(x - X_t), x - X_t \rangle + \frac{1}{2\gamma_t}\|x - X_t\|^2$

4: $\qquad X_{t+1} = \arg\min_{x \in \mathcal{X}} \langle a_t g(\bar{X}_{t+\frac{1}{2}}, \xi_{t+\frac{1}{2}}), x \rangle + \frac{1}{2\gamma_t}\|x - X_t\|^2$

5: **end for**

---

**Universal step-size**   We modify our step-size (see Eq. (2)) in order to operate in the stochastic regime while making it noise-adaptive for rate interpolation. Using the same weighted averaging scheme in Eq. (6), we define the universal counterpart of the adaptive step-size, Note that $\gamma_t$ is independent of any variable/randomness generated at iteration $t$; it accumulates $a_t^2 \|g(\bar{X}_{s+\frac{1}{2}}, \xi_{s+\frac{1}{2}}) - \tilde{\mathbf{F}}(\bar{X}_{s+\frac{1}{2}}; \tilde{X}_s, \xi_t)\|^2$ up to $t - 1$. Therefore, the step-size is decoupled from the explicit update, *a priori*.

Now, what remains is a new algorithmic design that will retain the accelerated convergence properties demonstrated by (Implicit) while having an explicit construction that is capable of automatically adjusting to noise level in the oracle feedback. Before expanding upon the technical details of our strategy, let us take our time to explain the consequences of our explicit design compared to (Implicit).

**From implicit to explicit**   To obtain the explicit algorithm, (*i*) we write the projection sub-problem in the $\arg\min$ form; (*ii*) introduce *stochastic* oracle feedback; (*iii*) for the second-order term, replace $X_{t+\frac{1}{2}}$ in $\bar{X}_{t+\frac{1}{2}}$ with the free variable $x$; then, (*iv*) simplify as follows:

$$\frac{a_t}{2}\langle H(\tilde{X}_t, \xi_t)(\bar{X}_{t+\frac{1}{2}} - \tilde{X}_t), x - X_t \rangle$$
$$\Downarrow$$
$$\frac{a_t}{2}\left\langle H(\tilde{X}_t, \xi_t)\left(\frac{b_t X_{t+\frac{1}{2}} + \sum_{s=1}^{t-1} b_s X_{s+\frac{1}{2}}}{B_t} - \frac{b_t X_t + \sum_{s=1}^{t-1} b_s X_{s+\frac{1}{2}}}{B_t}\right), x - X_t \right\rangle$$
$$\Downarrow$$
$$\frac{a_t}{2}\left\langle H(\tilde{X}_t, \xi_t)\left(\frac{b_t x + \sum_{s=1}^{t-1} b_s X_{s+\frac{1}{2}}}{B_t} - \frac{b_t X_t + \sum_{s=1}^{t-1} b_s X_{s+\frac{1}{2}}}{B_t}\right), x - X_t \right\rangle$$
$$\Downarrow$$
$$\frac{a_t b_t}{2B_t}\langle H(\tilde{X}_t, \xi_t)(x - X_t), x - X_t \rangle$$

Given the bisection-type conceptual methods [49, 30, 12], it is surprising how smoothly we could transition from implicit to explicit *once* we decouple the step-size from the current iteration *apriori*. Moreover, the resulting update rule for the extrapolation step retains the quadratic structure as the $X_{t+1}$ update rule. Having analyzed the components of the explicit scheme, we will first present the universal convergence rates then provide a concise explanation of the proof strategy with particular emphasis on the principal components of the analysis.

**Theorem 3.2.** *Let $\{X_{t+\frac{1}{2}}\}_{t=1}^{T}$ be a sequence generated by Algorithm 1, run with the adaptive step-size policy (2) and $a_t = t^2, b_t = t^p$ with $p \geq 2$. Assume that $f$ satisfies (H-smooth), and that Assumptions (2) hold. Then, the following universal guarantee holds:*

$$f(\bar{X}_{T+\frac{1}{2}}) - f(x^*) \leq O\left(\frac{\frac{D^2+\gamma^2}{\gamma}\sigma_g}{\sqrt{T}} + \frac{\frac{D^3+D\gamma^2}{\gamma}\sigma_H}{T^{3/2}} + \frac{\max\left\{L\frac{D^4+D\gamma^3}{\gamma}, \sqrt{\beta_0}\frac{D^2+\gamma^2}{\gamma}\right\}}{T^3}\right)$$

*When $\gamma = D$, we obtain the target rate $O\left(\frac{D\sigma_g}{\sqrt{T}} + \frac{D^2\sigma_H}{T^{3/2}} + \frac{\max\{LD^3, \sqrt{\beta_0}D\}}{T^3}\right)$.*

*Remark* 3.2. Similar to Theorem 3.1, EXTRA-NEWTON achieves the preceding convergence rate independent of the knowledge of problem parameters.

Compatible with the (EG)-based algorithmic design, our proof has the following main steps

1. We perform an *offline* regret analysis of Alg. 1 and show adaptive regret bounds - see Prop. 3.1.
2. We prove an anytime online-to-batch conversion framework, which generalizes that of Cutkosky [18], through decoupling iterate averaging from gradient weighting - see Theorem 3.3.
3. Combining the adaptive regret bound with the conversion theorem immediately implies *universal, accelerated* value convergence of $O(\frac{D\sigma_g}{\sqrt{T}} + \frac{D^2\sigma_H}{T^{3/2}} + \frac{\max\{LD^3, \sqrt{\beta_0}D\}}{T^3})$ - see Theorem 3.2.

Let us begin with clarifying what *offline regret* means for Algorithm 1. We define the (linear) regret considering the convention in both online learning [62, 18] and first-order acceleration literature [66, 35, 31]. We measure the performance of our decisions for the extrapolation sequence such that after playing $X_{t+\frac{1}{2}}$, our algorithm observes and suffers the linear (weighted) loss with respect to $a_t \nabla f(\bar{X}_{t+\frac{1}{2}})$. Hence, we define the regret as

$$\mathrm{R_T}(x) = \sum_{t=1}^{T} a_t \langle \nabla f(\bar{X}_{t+\frac{1}{2}}), X_{t+\frac{1}{2}} - x \rangle \tag{Reg}$$

where we run the algorithm for $T$ rounds. Next up, we provide our generalized conversion result.

**Theorem 3.3.** *Let $\mathrm{R_T}(x^*)$ denote the anytime regret for the decision sequence $\{X_{t+\frac{1}{2}}\}_{t=1}^{T}$ as in (Reg), and define two sequences of non-decreasing weights $a_t$ and $b_t$ such that $a_t, b_t \geq 1$. As long as $a_t/b_t$ is ensured to be non-increasing,*

$$f(\bar{X}_T) - f(x^*) \leq \frac{\mathrm{R_T}(x^*)}{a_T \frac{B_T}{b_T}}$$

*Remark* 3.3. This conversion result holds independent of the order of smoothness of the objective as long as $f$ is convex. Moreover, it allows averaging parameter $b_t$ to be asymptotically larger than gradient weights $a_t$, enabling a more aggressive averaging strategy when necessary.

To complement the lower bound to the regret $\mathrm{R_T}(x^*)$, we present an upper bound that helps us explain how we exploit second-order smoothness for a more aggressive weighting, hence the rate $O(1/T^3)$.

**Proposition 3.1.** *Let $\{X_{t+\frac{1}{2}}\}_{t=1}^{T}$ be generated by Algorithm 1, run with a non-increasing step-size sequence $\gamma_t$ and non-decreasing sequences of weights $a_t, b_t \geq 1$ such that $a_t/b_t$ is also non-increasing. Then, the following guarantee holds:*

$$\mathbb{E}\mathrm{R_T}(x^*) \leq \frac{1}{2}\mathbb{E}\left[\frac{3D^2}{\gamma_{T+1}} + \sum_{t=1}^{T}\gamma_{t+1}a_t^2\|g(\bar{X}_{t+\frac{1}{2}}, \xi_{t+\frac{1}{2}}) - \tilde{\mathbf{F}}(\bar{X}_{t+\frac{1}{2}}; \tilde{X}_t, \xi_t)\|^2 - \frac{\|X_{t+\frac{1}{2}} - X_t\|^2}{\gamma_{t+1}}\right]$$

Observe that the inequality in Proposition 3.1 is agnostic to the design of our step-size in Eq. (2) as well as the selection of the weights as described in Theorem 3.2. It essentially applies to any non-increasing sequence of step-sizes and non-decreasing gradient weight sequence $a_t \geq 1$. To obtain it, we neither used convexity nor the smoothness of the objective. In fact, the structure of the objective function, i.e., its convexity, will not be needed for upper-bounding the regret expression, and required only for the conversion in Theorem 3.3.

Now, let us explain how we make use of second-order smoothness for enjoying faster rates, and give a brief discussion of how the regret bound will look in its final form. First, we decompose the stochastic term $\|g(\bar{X}_{t+\frac{1}{2}}, \xi_{t+\frac{1}{2}}) - \tilde{\mathbf{F}}(\bar{X}_{t+\frac{1}{2}}; \tilde{X}_t, \xi_t)\|^2$ into deterministic feedback and noise. Then, we argue that *the noisy component* grows as $O(\sigma_H T^{3/2} + \sigma_g T^{5/2})$. On the other hand, achieving the accelerated $O(1/T^3)$ component of the universal rate amounts to showing that the regret has a constant, $O(1)$, component. In the worst-case sense, however, *the deterministic component itself* grows as $O(T^{5/2})$. Fortunately, we identify that the negative term is "large enough" in magnitude to control the growth of the deterministic term, permitting a constant component $O(LD^2)$ for the regret.

Although the regret bound of $O(LD^3 + D^2\sigma_H T^{3/2} + D\sigma_g T^{5/2})$ seems counter-intuitive from an online-learning perspective, it will make perfect sense when we discuss how second-order smoothness leads to "faster" conversion through more aggressive averaging. As a matter of fact, we will continue our discussion with how second-order smoothness helps us accelerate. It turns out that using (H-smooth), iterate averaging as in Eq.(6) and compactness of $\mathcal{X}$, we can bound the negative term as,

$$-\frac{1}{\gamma_{t+1}}\|X_{t+\frac{1}{2}} - X_t\|^2 \leq -\frac{1}{L^2 D^2 \gamma_{t+1}} t^4 \|\nabla f(\bar{X}_{t+\frac{1}{2}}) - \mathbf{F}(\bar{X}_{t+\frac{1}{2}}; \tilde{X}_t)\|^2$$

Observe that to seamlessly combine the positive and negative terms, our analysis enforces that $a_t = O(t^2)$ and $b_t = \Omega(t^2)$. Then, the conversion implies a convergence rate of $R_T(x^*)/T^3$, hence the recipe for acceleration. Therefore, the constant component of the regret amounts to $O(1/T^3)$ convergence rate, while the stochastic component of the regret implies $O(\sigma_H/T^{3/2} + \sigma_g/\sqrt{T})$ rate, giving us the first universal acceleration beyond first-order smoothness.

Let us conclude by discussing the intricate relationship between the universal step-size and the regret bounds. Simply put, growth of the summation in the denominator of $\gamma_t$ is of the same order as the regret bound. Under stochastic gradient and Hessian oracles, the regret bound is of order $O(T^{5/2})$, and we can trivially show using variance bounds that the step-size is lower bounded by $O(T^{-5/2})$. On the other extreme, the regret bound described in Proposition 3.1 is bounded by a constant under deterministic oracles, which implies that the summation in the denominator of the step-size is in turn summable, i.e., the step-size has a positive, constant lower bound. This adaptive behavior of our step-size enables automatic adaptation to noise levels and thus the universal rates.

## 4   Experiments

In this section, we will present practical performance of EXTRA-NEWTON against a set of first-order algorithms, e.g., GD, SGD, ADAGRAD [20], ACCELEGRAD [41], UNIXGRAD [35]; and second-order methods, e.g., NEWTON'S, Optimal Monteiro-Svaiter (OPTMS) [13], Cubic Regularization of Newton's method (CRN) [58] and Accelerated CRN (ACRN) [54] for least squares and logistic regression problems over a LIBSVM datasets, a1a and a9a. Our main objective is three-folds. First, when the objective has a favorable structure as in least squares, second-order method has cheap oracle costs and display superior convergence behavior. Second, we want to demonstrate the improved rates of our algorithm against accelerated and non-accelerated first-order methods through the $\ell_2$-regularized logistic regression problem. Finally, we compare our methods with respect to other second-order methods that achieve (almost) optimal rates. In the plots, the statement *# of oracle calls* on the x-axis counts any gradient or Hessian computation as one oracle call. Also note that we consider the black-box oracle model in which the algorithms only have access to gradient and Hessians without knowing the actual objective function.

When the problem is suitable, second-order methods show promising performance with truly superior run time. In Figure 1a, we display the result for least squares setting. Second-order methods are known to be suitable for quadratic problems, and our method exploits its hybrid construction to converge significantly faster than first-order methods, matching the behavior of NEWTON'S. For the logistic regression problem, we regularize it with $g(x) = 1/2\|x\|^2$, but use a very small regularization constant to render the problem ill-conditioned, making things slightly more difficult for the algorithms [47, 48]. Although we implement NEWTON'S with line-search, we actually observed a sporadic convergence behavior; when the initial point is close to the solution it converges similarly to EXTRA-NEWTON, however when we initialize further away it doesn't converge. This non-convergent behavior has been known for NEWTON'S, even with line-search present [27]. On the contrary, EXTRA-NEWTON consistently converges; even if we perturb the initial step-size and make it adversarially large, it manages to recover due to its adaptive step-size. We complement our numerical tests by comparing EXTRA-NEWTON with a set of second-order methods. To that end, we implemented our method within the framework presented in [13]. Using the implementation and the experimental setup provided in their GitHub repository [24], we implemented our method in their code and compared against NEWTON'S, CRN, ACRN and OPTMS algorithms. Figure 2 shows that EXTRA-NEWTON has comparable performance to OPTMS, which has the theoretically faster rate $O(1/T^{7/2})$, and marginally outperforms with respect to number of linear system solutions since the linesearch procedure of OPTMS might require multiple system solutions per iteration. While CRN and ACRN has worse convergence than EXTRA-NEWTON, NEWTON'S seems to have the fastest.

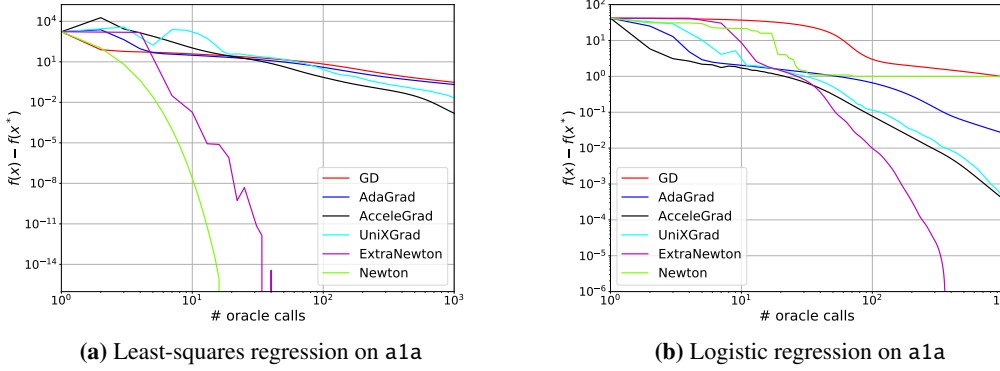

**(a)** Least-squares regression on `a1a`

**(b)** Logistic regression on `a1a`

**Figure 1:** Comparison of value convergence for regression problems with deterministic oracle access

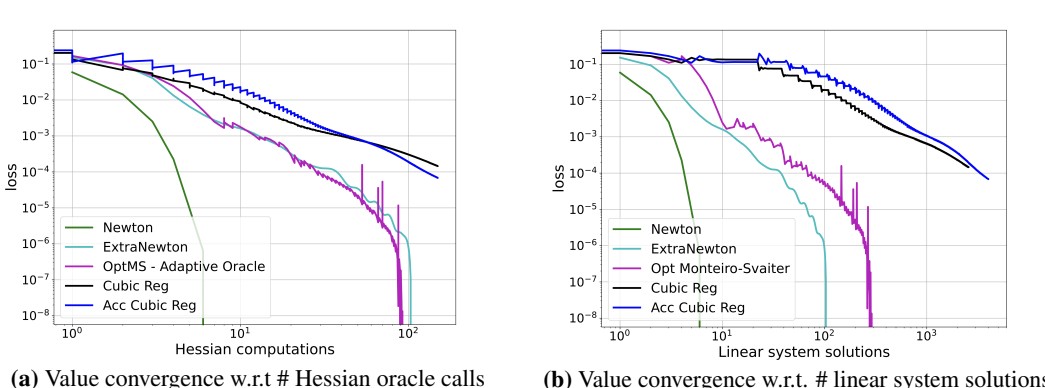

**(a)** Value convergence w.r.t # Hessian oracle calls

**(b)** Value convergence w.r.t. # linear system solutions

**Figure 2:** EXTRA-NEWTON vs. second-order methods. Logistic regression with `a9a` dataset

Note that the initialization favors NEWTON'S as it lies in a close neighborhood of the solution, and NEWTON'S performance sporadically deteriorates when initialized arbitrarily.

## 5 Conclusion

In this work, we present the *first* universal, second-order algorithm, EXTRA-NEWTON, which enjoys the value convergence rate of $O(\sigma_g/\sqrt{T} + \sigma_H/T^{3/2} + 1/T^3)$. By extending the notion of bounded variance on stochastic gradients to stochastic *Hessian*, we prove adaptation to the noise in first and second-order oracles, simultaneously, while showing accelerated rates matching that of Nesterov [54] under the fully deterministic oracle model. To that end, an important open question is whether we could design a method that achieves an improved rate interpolation guarantee $O(\sigma_g/\sqrt{T} + \sigma_H/T^{3/2} + 1/T^{7/2})$ without depending on any line-search/bisection mechanism. We defer this to a future work.

## Acknowledgments

This project has received funding from the European Research Council (ERC) under the European Union's Horizon 2020 research and innovation programme (grant agreement n° 725594 - time-data) and the Swiss National Science Foundation (SNSF) under grant number 200021_205011.

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
