# OpenReview forum: "Extra-Newton: A First Approach to Noise-Adaptive Accelerated Second-Order Methods"
_NeurIPS.cc/2022/Conference — NeurIPS 2022 Accept_

### Official Review · Reviewer_2XcJ · 2022-07-09

**Rating:** 8
**Confidence:** 3
**Soundness:** 4 excellent
**Presentation:** 3 good
**Contribution:** 4 excellent

**Summary:**

The authors propose an accelerated second-order method interpolating between global convergence rate of $O(\sigma/\sqrt{T})$ and $O(1/T^3)$ for minimizing Hessian-smooth, convex functions with stochastic / deterministic oracles, while being oblivious to knowledge of problem parameters. This is achieved by (1) extra-gradient framework incorporating second-order information; (2) careful iterate and gradients averaging for the updates that keeps memory of past iterates; (3) an adaptive step-size scheme that can automatically adapts to noisy feedback and bypass any potentially complicated line-search procedure needed for estimating relevant problem-dependent parameters. Numerical experiments are also presented showing speedup and stability compared to alternatives.

**Questions:**

Minor comments / typos: (1) it's a bit unclear what line 115 "the operator F satisfies eq (1)" means here; (2) after line 122 (EG): I assume you mean $x_t -> X_t, x_{t+1/2} -> X_{t+1/2}$? (3) Cubic-regularized Newton might be another nice addition to benchmark for the numerical section, considering it is also a second-order method that is both implementable and stable.

**Limitations:**

Yes.

**Strengths And Weaknesses:**

The paper is well-written and relevant work thoroughly surveyed. As a practical globally convergent second order method that is both universal and adaptive, it is of clear significance and relevance. In terms of the technical contribution, I find the construction of the policy for the adaptive step-size particularly interesting and similar ideas might find application in other second-order methods as well.

---

> ### Author Response · Authors · 2022-08-02
> **Initial Author Response**
>
> We would like to thank the reviewer for their encouraging comments and their positive evaluation. In what follows we shall address the particular comments and questions which the reviewer raised individually.
>
> 1. Equivalently this means that the objective $f$ possesses Lipschitz continuous Hessians. We will rephrase accordingly.
> 2. The reviewer is correct and we apologize. We will correct it in the revised version.
> 3. We already started implementing this comparison. We thank the reviewer for their suggestion. Please find our new set of experiments in the following link (https://imgur.com/a/kCUluEe).

---

### Official Review · Reviewer_fZEZ · 2022-07-10

**Rating:** 7
**Confidence:** 4
**Soundness:** 4 excellent
**Presentation:** 3 good
**Contribution:** 4 excellent

**Summary:**

In this work, the authors propose a new accelerated second-order method for
convex optimization. The convergence rate of this algorithm on the standard
class of convex functions with Lipschitz continuous second derivative is $O(1 /
T^3)$, which is exactly the same as that of the standard Accelerated Cubic
Newton Method. However, in contrast to the latter, the new method neither
requires the knowledge of the Lipschitz constant, nor uses any "line search"
for estimating it. Instead, the method accumulates certain quantities, that
measure the deviation of the gradients from their linear approximations, and
uses this information for computing the next iterate. The main feature of the
new method is that it can work not only with a deterministic oracle but also with a
stochastic one. Moreover, the method is "universal" in the sense that it
automatically adapts to the actual variance of the oracle, and therefore
requires no changes in its algorithmic scheme compared to the deterministic
case.

**Questions:**

Major remarks:

1. The notation $g(x)$ and $H(x)$ for stochastic gradients and Hessians is
   confusing as it does not highlight the randomness in the stochastic oracle.
   For example, it is not clear whether you use the same random variable for
   generating $g(\\tilde{X}\_t)$, $H(\\tilde{X}\_t)$ and $g(\\bar{X}\_{t + 1/2})$ in
   Algorithm 1, or different ones. A better notation would be $g(x, \\xi)$ and
   $H(x, \\xi)$, where $\\xi$ is a random variable from a certain distribution
   $\\mathcal{D}$. In this case, you can explicitly state in Algorithm 1 that you
   sample, e.g., $\\tilde{\\xi}\_t, \\bar{\\xi}\_{t + 1/2} \\sim \\mathcal{D}$, and then
   compute $\\tilde{g}\_t \= g(\\tilde{X}\_t, \\tilde{\\xi}\_t)$, $\\tilde{H}\_t \=
   H(\\tilde{X}\_t, \\tilde{\\xi}\_t)$ and $\\bar{g}\_{t + 1/2} \= g(\\bar{X}\_{t + 1/2},
   \\bar{\\xi}\_{t + 1/2})$.

2. I do not see the purpose of presenting the "implicit algorithm". To me, it
   looks rather confusing. Furthermore, as far as I understand, it is completely
   identical to Algorithm 1 (considering the deterministic oracle). If this is
   indeed the case, then what is the need for introducing such redundancy in the
   paper (1 extra page in the main part, and 6 extra pages in the supplementary
   material)? In my opinion, Algorithm 1 is sufficiently simple in itself.
   Therefore, I would suggest to completely remove the "implicit algorithm" (and
   its convergence analysis), and use the free space to better describe
   Algorithm 1 and its convergence analysis. To make the method easier to
   understand, instead of splitting it into "implicit" and "explicit" versions,
   it might be reasonable to first discuss the deterministic case and then
   quickly describe what becomes different in the stochastic case.

3. I suggest incorporating the updates of $\\tilde{x}\_t$, $\\bar{x}\_{t + 1 / 2}$,
   $\\gamma\_t$ and $B\_t$ directly into Algorithm 1 to make it easier to
   understand. Otherwise, it is not clear that there are no cyclic dependencies
   in the definitions.

4. Algorithm 1, line 2: It would be better to add some comments about the
   complexity of this step, as this is no longer the standard Euclidean
   projection. For example, in the important case, when the set $\\mathcal{X}$ is
   a Euclidean ball, this step can be efficiently computed using a tridiagonal
   factorization of the Hessian; the complexity is then similar to that of the
   standard Cubic Newton step. Another interesting case is when $H(\\tilde{X}\_t)$
   has low-rank structure.

5. Eq. (8): Why is there "$1 + \\ldots $"? Can we have there any other constant
   instead of $1$? The problem with $1$ is that it is a "wrong" constant, as it
   does not correctly interact with the squared gradient terms (they have
   different physical dimensions). More precisely, in its current form, eq. (8)
   makes Algorithm 1 not scale-invariant. Indeed, if we multiply the objective
   function by some positive constant $\\tau > 0$, then all the gradients and
   Hessians will also be multiplied by $\\tau$, but $\\gamma\_t$ will not be
   divided by $\\tau$, and therefore the method will construct a completely
   different trajectory $x\_{t + 1 / 2}$ (although the problem did not really
   change). Note that we would not have this issue if there was, e.g., "$0 +
   \\ldots$".

6. Theorem 3.2: There are some typos in the formulas. In any case, I suggest
   presenting the complete 3-term guarantee which appears between lines 769 and 770.
   It would be good to emphasize that the first- and second-order noise
   terms decrease at different rates.

7. Why do we take $b\_t \= t^3$ in Theorem 3.2? If I understand correctly, a
   simpler choice $b\_t \= t^2$ also works.

8. Theorem 3.3 and Proposition 3.1: As far as I understand, we do not need the
   part "non-decreasing weights $a\_t$ and $b\_t$ such that $a\_t, b\_t \\geq 1$".

9. Why do we not choose explicitly $a\_t \= (B\_t / b\_t)^2$? As far as I
   understand, this is exactly what is needed in the proofs (plus the fact that
   $a\_t / b\_t$ is decreasing, which can be easily ensured by taking $b\_t \= t^p$
   with $p \\geq 2$).

10. Experiments: 1) What $f$ and $\\mathcal{X}$ do you use in the experiments?
   Please specify them using formulas. 2) Please clarify whether the methods use
   the stochastic oracle (and which), or not. 3) What is the point in doing
   regression and classification on the same data set (a1a)? 4) Why does Newton
   method not convergence in one iteration in Figure 1a)? Is it not a quadratic
   problem? 5) It is surprising that ExtraNewton exhibits superlinear
   convergence, as it is not common for accelerated second-order methods to
   possess such a property (e.g., Accelerated Cubic Newton does not have
   superlinear convergence). Do you have any explanation for this? 6) It might
   be reasonable to add Cubic Newton and its accelerated variant to your
   comparison.

Minor remarks:

1. The convergence rate of the new method should be $L D^3 / T^3$, not $L D^2 /
   T^3$. There are multiple such typos throughout the paper, see, e.g., lines
   78, 247, 255, etc.

2. Line 78: Please clarify what the constant $L$ is.

3. Line 81: It is not exactly true that the method is oblivious to the diameter
   of the constraint set since the "correct" value of the parameter $\\gamma$
   depends on the diameter. Please clarify this. The same applies to the
   Abstract.

4. Lines 97--102: You should assume that you work either in an abstract
   $d$-dimensional Euclidean space $\\mathcal{V}$, or in $\\mathbb{R}^d$; there is
   no point in mixing them up by stating that $\\mathcal{V} \= \\mathbb{R}^d$.
   Further, since you work with a smooth function $f$, instead of "proper, lower
   semi-continuous, ..." and "Lipschitz continuous selection ...", you should
   simply assume that $f$ is twice differentiable (over some open set containing
   $\\mathcal{X}$). Finally, note that Assumption 2.1 is satisfied automatically
   since $\\mathcal{X}$ is a compact set.

5. Section 2: Please clarify that you use standard Euclidean and operator norms.

6. Eq. (Taylor): I suggest removing it.

7. Eq. (EG): There should be capital $X\_t$ and $X\_{t + 1/2}$.

8. Eq. (5): Some constants ($L$ and $D$) are missing.

9. Algorithm 1, line 2: The coefficient should be $\\frac{a\_t b\_t}{2 B\_t}$. Also,
   the definition of $B\_t$ is missing.

10. Eq. (8): I suggest denoting $\\| g(\\bar{X}\_{s + 1 / 2}) -
    \\tilde{F}(\\bar{X}\_{s + 1 / 2}; \\tilde{X}\_s) \\|$ somehow (e.g.,
    $\\tilde{\\Delta}\_s$), and then use $\\tilde{\\Delta}\_s$ everywhere instead of
    the long expression that appears in multiple places throughout the paper.

11. Theorem 3.2: The definition of $D$ is missing.

12. Theorem 3.3: There should be $\\bar{X}\_{T + 1 / 2}$ instead of $\\bar{X}\_T$.

13. Proposition 3.1: The parentheses are missing after the sum sign. Also, it
    might be reasonable to remove "Let $\\{ X\_{t + 1 / 2} \\}\_{t \= 1}^T$ be
    generated by Algorithm 1", or rephrase it.

14. Eq. (11): In the first line, there should be no "$2$" in the denominator.

15. Display between lines 617 and 618: Please simplify it, as everything
    trivially follows from the fact that $\\mathbb{E}[ g(\\bar{X}\_{t + 1 / 2}) |
    X\_{t + 1 / 2} ] \= \\nabla f(\\bar{X}\_{t + 1 / 2})$.

16. Lemma F.1: There should be $\\alpha\_T$ instead of $\\alpha\_t$. Also, one
    should additionally assume that $\\alpha\_1 > 0$.

17. Display between lines 732 and 733: Expectation sign is missing in the
    right-hand side.

18. Display between lines 735 and 736: It would be better to remove the negative
    signs and rewrite everything in terms of positive quantities (reversing the
    inequalities). Also, please specify what $c$ is.

19. Eq. (20): There should be $-\\delta\_t$ instead of $\\delta\_t$.

20. Display between lines 747 and 748: I suggest introducing some notation for
    $\\min\\{ \\| \\tilde{\\nabla}\_t \\|, \\| \\nabla\_t \\| \\}$ to avoid repeating the
    same long expression everywhere.

21. Display between lines 756 and 757: Please clarify why $\\sqrt{1 + \\sum\_{t \=
    1}^T a\_t^2 \\min \\ldots} \\leq \\gamma^{-1} \\sum\_{t \= 1}^T \\lambda\_{t + 1}
    a\_t^2 \\min \\ldots$.

22. Display between lines 759 and 760: Can we guarantee that $T\_0$ is always
    finite? If not, please add some comments about the case $T\_0 \= \\infty$.

23. Display between lines 769 and 770: Expectation sign is missing.

**Strengths And Weaknesses:**

This work presents a new approach for constructing accelerated second-order
methods. An unusual feature of this approach is that it does not rely on any
"line search" and can be easily extended to the case of the stochastic oracle.
Despite many similarities to the already existing acceleration techniques, the
construction, proposed in this work, definitely has a few novel elements.
Overall, I find this paper very interesting and believe that the ideas,
presented there, could be useful in a broader context.

The main drawback of the new method is that it works only with a bounded
feasible set. In particular, it cannot be directly applied for unconstrained
optimization problems. Also, the convergence rate $O(L D^3 / T^3)$ of the new
method depends on the diameter $D$ of the feasible set, in contrast to the $O(L
D\_0^3 / T^3)$ rate of the standard Accelerated Cubic Newton Method which depends
on the distance $D\_0$ from the initial point to the solution ($L$ is the
Lipschitz constant of the second derivative). Nevertheless, in my opinion, this
drawback is not really significant and can be addressed in future work.

Please see my detailed remarks in the next section. I will be happy to increase my score further if the authors address these remarks.

---

> ### Author Response · Authors · 2022-08-02
> **Initial Author Response (2/2)**
>
> **7.**
>
> We will write $b_t = t^p$ where $p \geq 2$ to cover all the possible cases. In our manuscript we specifically chose $b_t = t^3$ as this is the case we used in the experiments, too.
>
> &nbsp;
>
> **8.**
>
> $a_t, b_t$ should be non-decreasing to ensure that we could lower bound the error term that is generated when $a_t \neq b_t$. Otherwise, we cannot guarantee positivity of the quantity:
>
> $$\sum_{t=1}^{T-1} B_t \left( \frac{a_t}{b_t} - \frac{a_{t+1}}{b_{t+1}} \right) \left( f(\bar X_{t+1/2}) - f* \right). $$
>
> in the proof of Thm. 3.3. It must be positive for conversion theorem to check out when $b_t = t^p, p > 2$. We might be able to ensure this with a more complex rule over $a_t, b_t$ but we identify our condition to be the simplest.
>
> &nbsp;
>
> **9.**
>
> The reviewer's proposition would also be valid to achieve the same rates of conversion, this is a quite detailed observation. We wanted to highlight that acceleration beyond $O(1/T^2)$ requires a more aggressive averaging and weighting scheme than the existing literature proposes, which we try to explain at several places in the manuscript. We also wanted to identify that iterate averaging and gradient weighting could be decoupled and they could grow at different rates. Also for simplicity reasons, we chose to set $a_t = t^2$ and $b_t = t^p, p \geq 2$.
>
> &nbsp;
>
> **10.**
>
> We will formally write the logistic regression and least-squares losses in explicit form. We considered the unconstrained case since not all the methods have theoretical guarantees under constraints (e.g. AcceleGrad). Moreover, we wanted to explore this case at least experimentally since as mentioned the lack of theoretical guarantees is prevalent even for first order universal methods.
> When the setting is stochastic, we use stochastic (minibatch) oracles for both gradient and the Hessian. We will clearly state their construction for the stochastic setting.
>
> Perhaps we misunderstood the reviewer’s comment, but we only consider regression problems rather than classification as our main focus is the rate of decrease in training loss. Could the reviewer please explain their comment on classification?
>
> For one-iteration convergence of Newton’s method, we need to make sure the data matrix is invertible for least-squares. With real data, this is rarely the case hence we need to resort to an iterative scheme.
>
> We numerically observe that an adaptive version of the Monteiro-Svaiter algorithm also achieves some sort of a superlinear rate. We actually do not have a concrete explanation as the local convergence analysis requires fundamentally different tools. Still, we agree that this is surprising.
>
> Please find our new set of experiments comparing multiple second-order methods (https://imgur.com/a/kCUluEe)
>
> &nbsp;
>
> **Minor Remarks:**
>
> Once more we would like to thank the reviewer for their extremely detailed proof-reading of our manuscript. Of course all their suggestions will be included in the revised version. We briefly comment on some of their minor remarks.
> &nbsp;
>
> **1.** It is a typo and we sincerely apologize for that.
>
> &nbsp;
>
> **2.** $L$ here denotes the second order smoothness constant. We are sorry for any confusion.
>
> &nbsp;
>
> **3.** By oblivious we meant that a prior knowledge of the feasible domain’s diameter is not required in order to guarantee the convergence and the specific rates of our method. However, in order to optimize the particular constants which appear in the final upper bounds, the reviewer is correct that this prior knowledge is needed. Therefore, we agree that a more accurate description of this situation would be helpful and hence we shall incorporate this in our revised version.
>
> &nbsp;
>
> **4.** Trying to deal with differentiability and constraints perhaps complicated things. Thus, we agree with the reviewer and we shall adopt their suggestion.
>
> &nbsp;
>
> **5.-8.** Consider them done.
>
> &nbsp;
>
> **9.** $B_{t}$ denotes the sum of the $b_{t}$ sequence. We are sorry for any confusion.
>
> &nbsp;
>
> **10.-23.** Consider them done.

---

> ### Author Response · Authors · 2022-08-02
> **Initial Author Response (1/2)**
>
> We would like to thank the reviewer for their encouraging comments and their positive evaluation. In what follows we shall address the particular comments and questions which the reviewer raised individually.
>
> &nbsp;
>
> **Major Remarks:**
>
> **1.**
>
> We agree that the reviewer's suggestion would be the complete and precise way of addressing the randomness. We will incorporate the suggested notation and update the definitions accordingly. As a small note, oracle calls in the same iteration should be made with respect to the same random vectors $\xi_t$ to make sure smoothness inequalities check out.
>
> &nbsp;
>
> **2.**
>
> Our purpose for introducing the implicit method was to demonstrate in a more detailed manner the main ingredients which are needed in order to obtain-adopting the Extra-Gradient language- a sharper extrapolation step.
>
> So, we would like to respectfully disagree with the reviewer’s suggestion to totally remove it. That being said, we plan to shorten this introductory algorithmic scheme in order to shed more light on the actual explicit method. We also want to point out that the implicit scheme and Algorithm 1 are not identical under deterministic oracles. If we write the Euclidean projection in the open form, one might observe that the extrapolation step of the **implicit algorithm** is:
>
> $$ X_{t+1/2} = \arg \min_{x \in \mathcal X} \langle a_t \nabla f(\tilde X_t), x \rangle + \langle \nabla^2 f(\tilde X_t) (\bar X_{t+1/2} - \tilde X_t), x \rangle + \frac{1}{2 \gamma_t} \lVert \rVert^2 $$
>
> This sub-problem consists of a linear term (inner products) and a quadratic term (regularizer), while Algorithm 1 consists of 1 linear term and 2 quadratic terms.
>
> &nbsp;
>
> **3.**
>
> We are in agreement with the reviewer that writing the algorithm and definitions in the way they suggest seems to be a clearer presentation of our algorithm despite the fact that we ensure that there are no cyclic definitions; namely our step-size and its lagging behind summation precisely addresses that. We will define weights, normalizing constants and the step-size in the algorithm itself to avoid any confusion.
>
> &nbsp;
>
> **4.**
>
> We give a partial answer to this in our response to Reviewer 9RHq, bullet 1. Given the reviewer’s perspective, it is indeed an important part of our scheme since the complexity of this sub-problem is crucial given the structure of the set $\mathcal X$. We will expand upon your suggestion with respect to cubic regularization and dedicate a section for discussing its computational complexity along with a comparison to existing approaches.
>
> &nbsp;
>
> **5.**
>
> Our idea with putting “$1 + …$” is to address the lag-one-behind structure of the step-size and simply make sure the step-size is initialized at iteeration 1, mostly for the ease of the analysis. However, the reviewer’s technical comment is perfectly valid and seems to be supported also numerically.
>
> One common practice in adaptive optimization is to put some upper bounds on the gradient norms, i.e., G^2, for first-order problems instead of 1, however, this requires an additional assumption. One solution to this scaling issue is to define $\gamma_1 = \gamma$. Then, for $\gamma_t$ with $t \geq 2$, we have in the denominator $\sqrt{0 + …}$. This will change the constant in the final bound but it would enable the desired adaptation to scaling as the reviewer suggests.
>
> &nbsp;
>
> **6.**
>
> We are very pleased with the diligence of the reviewer and recognizing the original 3-term convergence rate. Since there are no concrete lower bounds established for this setting where the gradient and the Hessian oracles have bounded variance, we wanted to keep a simpler, intuitive bound as our main result. We could also convey the idea of universality easier with this simpler notation.
>
> However, the 3-term bound is of great interest from a theoretical point of view, at least for us. We will consider the reviewer’s suggestion and write the rates in the 3-term form for better exposition of individual adaptation to noise in gradient and Hessian oracles.

---

> ### Author Response · Authors · 2022-08-09
> **Re-evaluation of the score and possible final remarks**
>
> We would like to thank you once more for your detailed evaluation of our paper with a thorough check of the proofs. We tried to respond to your comments and remarks in a clear and honest manner with sound technical explanations.
>
> If you find our answers satisfactory and convincing, we would like to kindly ask you to reconsider your score for our paper. In case you have any further remarks, we will be happy to respond to them to the best of our knowledge.

---

> > ### Comment · Reviewer_fZEZ · 2022-08-09
> > **After rebuttal**
> >
> > Thanks for your answers. Please make sure to include all these comments in the
> > revised version of the paper.
> >
> > I have the following comments regarding my original major remarks and your
> > response to them:
> >
> > - **Remark 2.** I still do not quite understand why they are not equivalent.
> >   Yes, in the formula from your response (which is broken for some reason),
> >   there is a "long" linear term and a quadratic term. But the linear term
> >   depends on the new point $x\_{t + 1/2}$. In the explicit algorithm, there is
> >   one "short" linear term and two quadratic terms, but nothing depends on the
> >   new point $x\_{t + 1/2}$. What happens if you write the optimality conditions
> >   for each problem and compare them? I suppose they will be exactly the same, or
> >   am I wrong?
> >
> > - **Remark 5.** I suggest you simply replace "$1 + \\dots$" with "$\\Delta +
> >   \\dots$" for some constant $\\Delta$, and see how the resulting convergence
> >   bound depends on $\\Delta$. If you really want to, you can always select
> >   $\\Delta = 1$ in your experiments, but at least please make sure that the
> >   theory is carried out for a general $\\Delta$.
> >
> > - **Remark 8.** I agree with the formula you wrote. But you only need the
> >   monotonicity of $a\_t / b\_t$ to ensure that each term in the sum is nonnegative
> >   (you do not need the inequalities $a\_t, b\_t \\geq 1$, and it is not important
> >   whether $a\_t$ and $b\_t$ are themselves monotone or not).
> >
> > - **Remark 10.**
> >
> >     3) What I meant is that logistic regression model is used for solving
> >     classification problems (which have a binary output: yes/no), while least
> >     squares is used for solving regression problems (which have a continuous
> >     output: real number). The dataset "a1a" corresponds to a classification
> >     problem, so I do not understand why do you run a regression model on it. Of
> >     course, in principle, you can, but what is the point? Wouldn't it better to
> >     run least squares on a regression problem?
> >
> >     4) The pure Newton's method should always converge in one step on a
> >     quadratic function. After all, this is exactly what it does at each
> >     iteration---minimizes the quadratic model of the function (which coincides
> >     with the function itself when the function is quadratic).

---

> > > ### Author Response · Authors · 2022-08-09
> > > **Thank you for your acknowledgment**
> > >
> > > Thank you for your quick reply. We already started incorporating the changes suggested by you and the other reviewers. Here are our replies.
> > >
> > > - **Remark 2:** You are right, their optimality conditions are the same, which is why we keep the implicit scheme as an easy introduction. The implicit algorithm is presented in the same sense as some of the bisection-type algorithms in the literature and we try to emphasize the difference in our approach in selecting the step-size and next point, $x_{t+1/2}$. As you pointed out, the implicit and explicit algorithms have the fundamental difference that the explicit algorithm has no dependence on $x_{t+1/2}$, hence we have 2 quadratic terms. Algorithmically speaking, they are different; but in terms of analysis, the optimality conditions are the same.
> > >
> > > - **Remark 5:** We acknowledge your suggestion. Indeed, what we proposed in our answer was in the same sense as you described where we would have a free parameter $\Delta$ rather than 1. We believe we phrased our suggestion in a not so clear way. We will incorporate this idea in our paper.
> > >
> > > - **Remark 8:** We included monoticity of the sequence as they were needed in the proof of the other Theorems, but for the conversion theorem, we do not need to have the monotonicity of the sequence. We will make sure to state the necessary and sufficient conditions on $a_t$ and $b_t$ only, in the sense you described in your remark.
> > >
> > > - **Remark 10:** We included logistic regression as a relatively "difficult" problem as quadratic problems are favored by second-order methods. We acknowledge your point about classification vs regression.
> > >
> > > We consider the black-box optimization setting where the algorithms only have access to gradient and Hessian oracles. It means that they are indifferent to the particular structure of the problem, hence it does not find the solution in one iteration; it needs to iteratively solve the problem with a linesearch procedure. We would like to additionally point out that even with a1a dataset, the matrix inversion raised numerical errors.
> > >
> > > In case you would be interested in further numerical results, we also had a new set of experiments for which we have a separate post in the general thread with links to the new figures.

---

> > > > ### Comment · Reviewer_fZEZ · 2022-08-09
> > > > **Additional comment**
> > > >
> > > > Dear authors,
> > > >
> > > > Regarding **Remark 2**, I am glad that we have come to an agreement. I do not
> > > > mind if you decide to keep the "implicit algorithm" in the paper (although, as I
> > > > already wrote in my initial review, I personally find it a bit confusing).
> > > > However, please make sure there no repetitions in the corresponding proofs.
> > > > Since the optimality conditions are the same, it suffices to carry out all the
> > > > proofs only once (and not twice, as was in the original version).
> > > >
> > > > Regarding **Remark 10**, note that you can always add a small regularization to
> > > > make the problem nondegenerate. However, I would suggest to completely remove
> > > > "least squares" from your experiments, as quadratic problems are indeed too
> > > > simple for second-order methods.

---

> > > > > ### Author Response · Authors · 2022-08-09
> > > > > **Additional response**
> > > > >
> > > > > Thank you once more for the quick response.
> > > > >
> > > > > - We will organize the proofs for implicit and explicit algorithm to avoid repetitions pertaining to the iterate updates with the same optimality condition (this mainly concerns Proposition 3.1 and Proposition G.1 for the proof of template regret bounds).
> > > > >
> > > > > - We were already planning to include the new set of experiments in the main text to provide a broader numerical evaluation on multiple fronts. We could also move the least-squares experiments to the appendix and put more emphasis on the logistic regression problems in the main text.
> > > > >
> > > > > In the light of our latest discussion, do you find our responses convincing such that you would reconsider your scoring?

---

> > > > > > ### Comment · Reviewer_fZEZ · 2022-08-09
> > > > > > **Response to Additional response**
> > > > > >
> > > > > > Thanks. Yes, I do find your responses convincing. I will maintain my original score.

---

### Official Review · Reviewer_9RHq · 2022-07-11

**Rating:** 4
**Confidence:** 4
**Soundness:** 2 fair
**Presentation:** 3 good
**Contribution:** 2 fair

**Summary:**

This manuscript proposes an algorithm for constrained smooth optimization that achieves simultaneously $O(1 / \sqrt{T})$ convergence rate for stochastic gradient and Hessian estimators and $O(1/T^3)$ convergence rate in the deterministic setting. The keys are a combination of the extragradient and the Newton method as well as a carefully chosen adaptive step size scheme.
The general idea is interesting and it leverages techniques from online learning/optimization so the context is indeed relevant to machine learning.
The manuscript is well-motivated with a clear explanation of its derivation, but the idea of adding a constraint set actually makes the algorithm non-implementable in practice, and thus, although interesting in theory, I doubt the usefulness of the proposed method.
There are also some unclear technical definitions in the theorems that need to be clarified. See the detailed comments.
Another concern is that the amount of experiments is far from sufficient to sufficiently support the performance of the proposed algorithm. I also have some concerns with the experimental results.

**Questions:**

1. My biggest concern is about line 2 of Alg. 1. When there is a constraint, clearly this subproblem is in general difficult to solve, especially since H(X) is in general nondiagonal. Thus it looks to me that this algorithm is still not implementable, just like the conceptual one in sec 3.1.

2. Another issue is surely the applicability of this algorithm and how to do a fair evaluation. The authors selected only low-dimensional problems, surely because the cost to invert the Hessian will otherwise be too high. In a more practical setting, usually the Newton step is obtained through an iterative algorithm like CG that queries the 2nd-order oracle n times (assuming that the exact Newton step is needed), where n is the problem dimension. I wonder how the convergence figure would change in that setting.

3. I also would like to know about the running time comparison of these algorithms, especially in the setting of higher-dimensional problems.

4. I am very confused about the results of Newton for least-square. Shouldn't the classical Newton's method take only 1 iteration to find the optimal solution? (And this also shows how this oracle call evaluation for least square is pointless.)

5. Thm 3.1 mentioned $\sigma$ in the assumptions, but the bound seems to be irrelevant to any stochasticity or expectation. I am thus wondering if there are any typos or missing things.

6. I don't quite see where the 1st equation after L. 236 is from. What does it mean and why out of nowhere this thing "to be simplified" appears.

7. Starting from Alg. 1, there suddenly is a quantity $B_t$ that is never defined in the text. What exactly is this?

8. Surely it is appealing theoretically to have a universal algorithm for all settings, but in what real-world application does one face such a situation such that we don't know whether the oracle is noisy or not?

9. This is not directly relevant to the evaluation of this work, but I am also curious about what would happen if the variance changes over iterations. Would it be possible to get a better rate in terms of the sum or so of the iteration-wise variances?

10. After reading others' reviews, I realized that I previously missed the point that this method does not apply to unconstrained optimization, and thus my first comment becomes vital, so I am lowering the evaluation. Also D is not defined in the main paper.

---
I am happy that the authors are willing to make the assumption on the constraint set explicit, so I'm increasing my score to 4.

**Limitations:**

As mentioned in the comments, the authors did not mention at all that solving a constrained second-order optimization problem to optimality is actually something difficult and thus also not implementable, just as the existing work they criticized.

They then went on to ignore this fact and conducted experiments on unconstrained problems, which is deceptive.

**Strengths And Weaknesses:**

Strength:
1. The work is well-motivated with a detailed explanation of its derivation.
2. The result seems to be novel up to my knowledge and of certain importance.

Weakness:
1. Insufficient numerical results
2. The algorithm is nonimplementable
3. There are key symbols used in the text that are undefined and hindered understanding of the results

---

> ### Author Response · Authors · 2022-08-02
> **Initial Author Response (2/2)**
>
> **4.**
>
> We understand the reviewer’s point, but note that we consider the black-box model in the experiments; algorithms have access to first and second-order oracles and the algorithm is not aware of the structure of the objective for least squares. Hence, we implement Newton’s method with linesearch, a common practice in black-box optimization, which naturally takes multiple iterations to converge. We will make this clear in the experiments section.
>
> Moreover, Newton’s method could find the solution with a single step **if and only if** the data matrix is invertible (or full rank). Most of the time, we cannot guarantee this, and in the real data regime, this is very rarely the case. Hence, one might need to solve the linear system inexactly and follow an iterative scheme.
>
> As for the divergence, it is a well-known fact that Newton’s method is not guaranteed to converge if it is initialized far away from the solution. Moreover, Newton’s method is sensitive to initializations; it might diverge under very small perturbations of a convergent initial point. Our experiments exemplify this phenomenon. In the additional experiments we provide, we especially initialize methods close to the solution to make sure Newton’s method converges.
>
> &nbsp;
>
> **5.**
>
> Your observation is correct, this is a typo and we will remove it in the final version.
>
> &nbsp;
>
> **6.**
>
> This equation is the result of the steps described in the above paragraph through (i), (ii), etc. It is simply the open form of the Euclidean projection. We will make it clear with all the steps in the final version.
>
> &nbsp;
>
> **7.**
>
> $B_t = \sum b_t$, which is the normalizing factor for the averaging weights. Unfortunately, we forgot to define it in the main text; however we have an exhaustive list of definitions in the appendix. We will make sure we define all the symbols and expressions before we use them in the main text.
>
> &nbsp;
>
> **8.**
>
> First, we want to highlight that the properties of universality (agnostic to the oracle’s type, i.e. deterministic/stochastic) and adaptivity (agnostic to the problem’s parameters) are testaments to the key-feature of **robustness** of our method; namely the performance of the method remains more or less unaffected under perturbations or errors. This feature is crucial throughout the ML literature. Therefore, obtaining these features are always of independent interest.
>
> For their particular question we refer the reviewer to the potential game literature introduced by (Shapley (1996)). Since each agent only knows their individual utility function, they do not know a priori whether they are facing a deterministic or stochastic minimization problem.
>
> &nbsp;
>
> **9.**
>
> We find your question very interesting and elaborate. Theoretically speaking, let us define the variance at every iteration as $\sigma_t^2$. If the variances satisfy $\sum_{t=1}^T \sigma_t^2 = O(T^p)$, where $p \in [0,1)$, then we could talk about a speed up in our rates. To give  specific example, if the variances are summable, i.e., $\sum_{t=1}^T \sigma_t^2 = O(1)$, then we would expect to have the fast convergence rate of $O(1/T^3)$. We would like to share our new plots for the low variance regime here (https://imgur.com/a/Ke0L2j8).
>
> &nbsp;
>
> **10.**
>
> We will define $D$ in the main text, thank you for pointing it out. The reviewer’s first comment is already answered above.
>
> &nbsp;
>
> **Limitations:**
>
> - *As mentioned in the comments, the authors …*:
>
> To clarify it once more, as we described to the reviewer’s first question, the projection subproblem in Alg. 1 is efficiently solvable via simple first-order subsolvers.
>
> Concerning the existing work, we only reiterated Nesterov’s comment (as cited in the footnote of Table 1)  that bisection-type methods achieve the optimal rates at the expense of running an iterative subsolver over an implicit scheme, which **makes extra second-order oracle calls** and **solves linear systems** every iteration of the sub-process. We never claimed that previous work has non-implementable subproblems in the sense the reviewer describes.
>
> For our experimental set-up, the reasons for our choice are the following:
> - Aiming to compare with the existing literature of universal first order methods, we wanted to conduct a fair comparison with the AcceleGrad method which cannot handle constraints (at least theoretically).
>
> - Providing empirical evidence for unconstrained problems for which even first-order universal methods do not exhibit any theoretical guarantees.
>
> &nbsp;
>
> *We hope that given our clarifications, you will be willing to change your evaluation, as we certainly do not think the reduced score fairly evaluates the new theory contributions for adaptive and universal second-order methods in our work.*

---

> > ### Comment · Reviewer_9RHq · 2022-08-07
> > **Response (1/2)**
> >
> > I thank the authors for their responses to my comments, although I am still not very satisfied with their replies.
> > Here are the major issues that remain for me:
> >
> > 1. Indeed from the oracle point of view there's no additional cost, but it doesn't mean at all that without an additional oracle call the constrained subproblem becomes easy. In particular, in L. 65, the authors criticized some existing methods for being prohibitively expensive computationally, but this is surely also the case in general for the proposed method.
> > Indeed there are some simple constraint sets such as the box constraints or the norm constraint for which the projection can be conducted easily, and in that situation, the authors' claim could be valid. Unfortunately, the paper boasted that they only need an assumption that the constraint is convex and compact, without any mentioning of the projection/proximal operation.
> > In many situations, even if X is convex and compact, projection onto it could be extremely costly, and thus the proposed method **in general** is still very expensive, while the authors only suggested **special cases** in response to my question.
> > And this is also very deceptive that the authors claimed all sort of contributions as if they have really solved the question in L. 38-39, but in fact they can only handle a very small subclass of problems (constrained, with the constraint set compact and convex, **and with the constraint set projection easy to compute** generally being considered ), while by being deceptive again to conduct experiments in a setting not covered by their theory, this seems to me to mislead people about the contributions of the theoretical part.
> > The authors should make this clear from the beginning, and then the degree of contribution of the new method should surely be reevaluated by all reviewers.
> >
> > 2. I don't see why a stochastic setting here matters: in your experiment you have Newton's method as a baseline for comparison, and what I am saying is simply replacing the operation of computing the exact inverse for solving the linear system (which has nothing to do with whether this is in a stochastic setting or not) with CG.
> > But surely here given a constraint set, (linear) CG is not applicable anymore to the proposed algorithm, so it will get a much higher cost for solving high-dimensional problems over other second-order methods that solve an unconstrained quadratic subproblem.
> > This is also closely related to the 1st point that in many cases, the proposed algorithm is still prohibitively expensive to be useful.
> > If there are restrictions/limitations of the proposed algorithm, the authors should have been clear about that up front.
> >
> > 3. Let me repeat my original question: I also would like to know about the running time comparison of these algorithms, **especially in the setting of higher-dimensional problems.**
> > The authors' response is still in this low-dimensional dataset that has d=123.
> >
> > 4. I was only talking about the **least-square case** in the **deterministic setting**, namely Fig 1(a). If you solved the subproblem (which is actually the original problem) exactly, any sort of reasonable linesearch procedure will accept unit step size in this case, and surely the initial point doesn't matter, because the solution provides the lowest possible objective.
> > Given the number of data points and the dimensionality of the problem, it is totally not what the authors described. Instead, Given that #instances >> #dimension, it is unlikely the Hessian is only positive semidefinite (which requires all data instances lie in a subspace whose dimension is smaller than the whole problem dimension).
> >
> > 9.The result sounds interesting, although I don't quite understand if the plot really shows what the authors described. But thanks for the extra check for the new scheme.

---

> > > ### Comment · Reviewer_9RHq · 2022-08-07
> > > **Response (2/2)**
> > >
> > > The part of limitations is also not quite satisfactory.
> > > Sorry for my inaccurate wording, I should have said prohibitive computation, and that way things could have been clearer.
> > > Here the issues are as discussed above, your subproblem is much harder to solve, and what you cited from Nesterov is under the premise (which he actually made but the authors here totally ignore) that the constraint set is a "simple" one. Note that the difference between this work and Accelegrad is that their algorithm takes the set K as something that can be self-defined, and thus in their case the cost of projection is not a problem because one could select a set that is simple, but it is not the case here.
> > > And although Accelegrad is for smooth optimization, surely if you select the set K as the constraint set X given in this work, everything will still apply, and thus I wouldn't think the authors' argument against that work is valid.
> > > Also, I don't think there's anyway to compare with AcceleGrad fairly: that method deals with unconstrained problems only (as the authors claimed, although this is not true as argued above) while the proposed one deals with constrained problems only.
> > >
> > > I therefore keep my score.

---

> ### Author Response · Authors · 2022-08-02
> **Initial Author Response (1/2)**
>
> We thank the reviewer for their comments. In what follows we address their concerns individually.
>
> &nbsp;
>
> **Questions:**
>
> &nbsp;
>
>
> **1.**
>
> We respectfully disagree with the statement for reasons that we explain below.
> First of all, we hope that the following is clear to the reviewer: We claim that the conceptual algorithm is not implementable because it requires an input that is not available. More precisely, the implicit method relies on a priori knowledge of the **next iterate** $X_{t+1/2}$ **before this is computed**, i.e. $X_{t+1/2}$ terms appear on both sides of the subproblem. Hence, our implicit method is not implementable in the same sense as  the **proximal point method**.
>
> Now, while “all optimization problems are unsolvable” (Nesterov, Introduction to Convex Optimization), when all inputs are given, some of them are indeed approximately solvable. Let us describe step by step how to generally solve the subproblem in line 2 of Alg. 1:
>
> $$ X_{t+1/2} = \arg \min_{x \in \mathcal X} \phi_t(x) := \langle a_t g(\tilde X_t), x  \rangle + \frac{a_t b_t}{2 B_t} \langle H(\tilde X_t)(x - X_t), x - X_t \rangle + \frac{1}{\gamma_t} \lVert x - X_t \rVert^2. $$
>
> In particular at any iteration $t$, the above update rule suggests that we call the gradient and Hessian oracles and compute $g(\tilde X_t)$ and $H(\tilde X_t)$ **once**. Now, having these at hand, we can construct the auxiliary function:
>
> $$\phi_t(x)=\langle a_t g(\tilde X_t), x  \rangle + \frac{a_T b_t}{2 B_t} \langle H(\tilde X_t)(x - X_t), x - X_t \rangle + \frac{1}{\gamma_t} \lVert x - X_t \rVert^2. $$
>
> Hence, in turn, we face the following minimization problem composed of linear and quadratic terms:
>
> $$\min_{x\in \mathcal{X}}\phi_{t}(x)$$
>
> A crucial additional observation here is that the $X_{t}$-dependent parts,  $g(\tilde X_t)$ and $H(\tilde X_t)$ are already computed and therefore can be treated as **constants** for the above minimization sub-problem.
> Therefore, having established the above setting, one could easily show that $\phi_t(x)$ is a smooth and strongly convex function, with explicitly known parameters.
>
> In turn, applying an inner loop (e.g., a simple projected gradient descent would suffice) we are able to solve this subproblem to machine precision within logarithmic number of steps; following the same reasoning as in Nesterov’s inexact methods (e.g. Nesterov (2014)) we can choose the required accuracy $\epsilon$ in such a way that the inexact solution is good enough and the quality of the candidate solution of our sub-problem is not affected.
>
> So, summarizing we make the following comments:
>
> - Throughout this inner loop we do not make **any further oracle calls** and a fortiori do not do any other prohibitive computations such as matrix inversions. Hence, in the worst case scenario this complexity incurs an additional **constant** logarithmic term.
>
> - Therefore, the difficulty of solving this sub-problem boils down to the computation of first-order projections and hence (up to logarithmic factors) the same complexity as in first-order projections applies.
>
> In the light of these clarifications, could the reviewer please clarify what they mean by “not implementable”?
>
> &nbsp;
>
> **2.**
>
> We believe there is a mismatch in expectations from the reviewer.
>
> To our knowledge, we provide the first theory for a universal Newton type of method with fast global convergence without linesearch. The numerical evidence is provided to show that the algorithm matches the rate and is robust to noise. Surprisingly, the evidence suggests that the algorithm  attains a superlinear rate around the solution in practice, which could hint at additional interesting settings for efficiency.
>
> Now, the recent work revolving around the Newton method also uses the same datasets from LibSVM library as a baseline [Mischenko, 2022, Doikov \& Nesterov 2020]. As the reviewer points out, inversion/linear system solutions have additional cost with increasing dimensions, but in this setting it is only fair to compare second-order methods amongst each other. To exemplify, we provide a new set of experiments, where we use a  larger dataset (with the same dimensions) in the problem setting of [Carmon et al., 2022].
>
> Could the reviewer please clarify their idea on the use of conjugate gradients in the stochastic setting?
>
> &nbsp;
>
> **3.**
>
> We observe that under run time plots, second-order methods are still dominant for least-squares (and in general for quadratic problems) as they are especially suited for such problems. For logistic regression, first-order methods have a matching performance with second-order methods. Please access our plots from here (https://imgur.com/a/bzUaCqj).

---

> ### Author Response · Authors · 2022-08-09
> **Possible final remarks**
>
> We thank the reviewer for reconsidering their evaluation of our paper. However, we feel that the reviewer has still unresolved concerns relative to our work despite the lenghty answers and our fairly detailed explanations.
>
> We strongly believe we answered all the concerns raised by our reviewer with diligence and utmost attention. Should the reviewer have any further concerns, we are willing to continue our discussions in a positive manner as in their initial review.

---

### Official Review · Reviewer_o7wn · 2022-07-13

**Rating:** 7
**Confidence:** 4
**Soundness:** 4 excellent
**Presentation:** 4 excellent
**Contribution:** 4 excellent

**Summary:**

The paper proposes a second-order method for minimizing second-order smooth and convex problems, based on an extra-gradient template (thus term it as ExtraNewton). The key feature of this method is that it uses a new stepsize rule that is independent of the smoothness modulus, variance bounds and the diameter of the compact set, which, as the authors claimed, makes the proposed method adaptive and universal. By introducing an optimistic weighting strategy for the decision variables as well as the gradients, they show that the proposed method can enjoy an accelerated rate of O(\sigma/\sqrt{T} +1/T^3) for a stochastic oracle so long as the sequence of weights are properly chosen and the above stepsize rule is employed. Numerical experiments have illustrated the performance of the proposed method in comparison to several existing well-known methods.

**Questions:**

- Will it be possible to remove the assumption of a set being compact, and what the rate will become if the assumption is void.
- How does the performance of the proposed method compared to those accelerated methods using cubic regularization for deterministic cases. The reviewer did not see such comparison in this paper.
- Except for the choice of a_t and b_t in this work, is there any other choice that may leads to better convergence performance (numerically)?
- In Figure 3, it can be observed that AcceleGrad outperforms others, including ExtraNewton in both least-square and logistic regression problems; is there any proper explanation on this since it should be better according to the theorectical rate obtained for the stochastic oralce?
- Since the algorithm employs second-order information, the reviewer would like to know (just for curiosity) if it is scalable to high-dimensional problems and if the analysis can be extended to stochastic problems with finite sum structure involving a great number of samples?


**Limitations:**

- The authors are suggested to make comparison (at least via numerical experiments) to the closely related accelerated method such as those relying on cubic regularization.
- The part of transiting implicit algorithm to explicit one (line 234- line240) lack of some intuitition and the authors are suggested to elaborate on this.
- In the statement of theorem 3.1 and 3.2, the reviewer guess that the Taylor condition and H-smoothness are both need to ensure (1), which most of the proofs rely on. Besides, the proofs for these above two theorems share a lot of similarity, thus it is suggested to, if possible, properly combine them.
- It would be of great interest if the authors can remove the assumption of compact sets and extend their analysis, if possible, to account for finite-sum problems.



**Strengths And Weaknesses:**

Strengths:
- The algorithm is new to the reviewer and it is a very nice property that the stepsize can be adaptive to the smoothness modulus which is usually hard to estimate. In addition, the algorithm is shown to enjoy a rate of O(\sigma/\sqrt{T}+1/T^3) for a stochastic oracle, which is not obtained before for existing second-order methods to the best of the reviewer’s knowledge.
- The paper is technically sound and some proof techniques seems interesting. The reviewer have checked most of the proofs which seems correct.
- The paper is well written and organized in a way that is very easy to follow; and the authors have demonstrated an excellent understanding of the existing literature on second-order optimization.


Weakness
- In the proof for the main results, the diameter of the compact set “D” seems very important to obtain the O(1/T^3) rate. However, this is quite restrictive for smooth problems, especially when the Hessian is also smooth.
- It follows from Fig. 3 that numerical performance of the proposed method seems not working very well when it comes to stochastic problems.
- The design of the adaptive stepsize rule lacks some intuition.

---

> ### Author Response · Authors · 2022-08-02
> **Initial Author Response (3/3)**
>
> **Limitations:**
>
> - *The authors are suggested to make comparison …*:
>
>     Please refer to the plots here (https://imgur.com/a/kCUluEe).
>
> - *The part of transiting implicit algorithm to explicit …*:
>
>     We will provide further explanations and derivations to make our perspective clear.
>
> - *In the statement of theorem 3.1 and 3.2, t…*:
>
>     We appreciate the reviewer’s elaborate evaluation of the proofs. We will try to present our results in a more concise way in the revised version.
>
> - *It would be of great interest if the authors …*:
>
>     As we already mentioned, we are in total agreement with the reviewer, and we are currently working towards this open problem. Finite-sum structure is another independent direction we will consider in future work.

---

> ### Author Response · Authors · 2022-08-02
> **Initial Author Response (2/3)**
>
> **Questions:**
>
> - *Will it be possible to remove the assumption of a set being compact …*:
>
>     We have to acknowledge that it is a difficult challenge; namely to show rate interpolation guarantees for unconstrained and/or unbounded domains with universal algorithms.
>
>     This difficulty is prevalent throughout the (primal-primal) first order minimization literature, where to the best of our knowledge, the only peer-reviewed paper that achieves universal acceleration for unconstrained problems is that of AcceleGrad (Levy et al (2018)). That being said, AcceleGrad analysis has a hidden **boundedness** requirement since it is assumed that the optimizer knows a priori a **domain of interest** where the global minimizer lies. A further intuition for this difficulty stems from the online convex optimization literature, where it is shown that boundedness of the domain is indispensable in order to obtain $\mathcal{O}(1/\sqrt{T})$ regret guarantee (Orabona \& Pal (2015)).
>
>     Therefore an interesting direction upon which we are working is to change gears to a primal-dual template for our candidate algorithm, building on the ideas of Dual Extrapolation scheme (see Nesterov (2007)) and combining it with the current analysis. With such an approach we would predict the algorithm also will provide theoretical guarantees for unconstrained and/or unbounded domains.  However, we defer this to future work.
>
> - *How does the performance of the proposed method compare …*:
>
>     We conducted an additional set of experiments under deterministic oracles where we considered the cubic regularization [Nesterov \& Polyak (2006)] and its accelerated counterparts.  We present convergence with respect to the number of Hessian oracle calls and also to the number of linear system solutions.
>
> - *Except for the choice of a_t and b_t in this work, …*
>
>     In a nutshell, what we observed in practice concerning the choice of $a_{t},b_{t}$ is that we need $a_t = t^2$ precisely. For $b_t = t^p$, we have flexibility to choose $p \geq 2$. We observe the best performance when $p \in [2,3]$.
>
> - *In Figure 3, it can be observed that …*:
>
>     Our main explanation for ExtraNewton has mainly to do with our step-size initialization. Namely, in the illustrated experiments we initialized the ExtraNewton’s step-size at $\gamma_{0}=1$. However, this initialization seems not to be optimal in practice. On the other hand, AcceleGrad’s step-size mechanism consists only of accumulating gradient norms (since it is originally constructed for unconstrained problems) and therefore can be randomly initialized.
>
>     However, our framework is actually open to tuning and remedy this situation. Please also refer to the following plots for the behavior under low variance regime (https://imgur.com/a/Ke0L2j8).
>
> - *Since the algorithm employs second-order information, …*:
>
>     Scalability is always a concern when it comes to second-order methods as they need to solve larger linear systems per iteration. Hence, ExtraNewton shows a similar behavior as any other second-order method does, and hence seems to be more appropriate for medium dimensional problems. Note that for quadratic problems, this is not an issue as the Hessian is a constant since one computes it once for the whole run.
>
>     As for the finite-sum structure, it seems that we should be able to extend our analysis, but obtaining optimal $n$ dependence (number of component functions) is not immediate. We find it an interesting direction as this is what one would observe in practice.
>
>     We will consider this a part of future extensions of our work.

---

> ### Author Response · Authors · 2022-08-02
> **Initial Author Response (1/3)**
>
> We would like to thank the reviewer for their encouraging comments and their positive evaluation. In what follows we shall address the particular comments and questions which the reviewer raised individually.
>
> **Weaknesses**
>
> - *In the proof for the main results, ...*:
>
>     In our manuscript, we originally concentrated on constrained problems with compact domains, as opposed to bounded iterates assumptions.
>     Would the reviewer be so kind as to clarify what they mean by restrictive?
>     For the particular issue of the diameter we shall elaborate in what follows.
>
> - *It follows from Fig. 3 …*:
>
>     While our motivation is mainly theoretical and tries to establish a new notion of convergence characterization in the study of second-order methods,  the experiments  show that we have comparable performance with their first order accelerated counterparts. To that end, we should highlight that $\mathcal{O}(1/\sqrt{T})$ is something to be expected at least theoretically due to the stochasticity of the first order information.
>     For the particular case of AcceleGrad, we shall elaborate in what follows.
>
> - *The design of the adaptive stepsize rule lacks some intuition.*
>
>     In order to address the reviewer’s concern we start by referring to the core references which inspired our construction:
>     1. The idea of accumulation of differences of gradients and Hessians due to [Rakhlin and Sridharan, 2013],
>     2. Incorporating aggressive averaging acceleration mechanisms to the step-size due to [Wang et al., 2018, Levy et al., 2018].
>
>     These two key ingredients result in our adaptive step-size in Eq. (7). Intuitively,  the machinery that allows the desired rate interpolation between the deterministic and the stochastic regime would the following:
>     From a simplified point of view, the growth of the upper bound of the inverse step-size (or lower bound on the step-size itself) is an indicator of the respective convergence rate.
>     For the stochastic case our step-size is lower bounded by $O(\frac{1}{T^{5/2}})$ (due to the bounded variance statistical assumption), which results in the $\mathcal{O}(\frac{1}{\sqrt{T}})$ rate via the conversion by Thm 3.3.
>     For the deterministic case, our step-size can be shown to stabilize at some (strictly) positive value $\gamma_{\infty}$, i.e., it converges to the said value. This stabilization results in a constant regret in the (RHS) inequality of Prop. 3.1. Finally, the improved rate is obtained by the online to offline conversion scheme in Thm 3.3.
>     We will be happy to incorporate this intuition in our camera ready of our manuscript.

---

### Author Response · Authors · 2022-08-02
**New experimental results**

To complement our answers to all the reviewers, we would like to bring our new set of plots to your attention:

- Timing plots for deterministic setting: https://imgur.com/a/bzUaCqj.
- Additional comparison with second-order methods: https://imgur.com/a/kCUluEe.
- Stochastic experiments in the lower variance regime: https://imgur.com/a/Ke0L2j8.

We will update this post when we add new plots and results, as per reviewers' request.

---

### Author Response · Authors · 2022-08-08
**A global discussion is requested (3/3)**

- **9QHq:** *The part of limitations is also not quite satisfactory. Sorry for my inaccurate wording, I should have said prohibitive computation, and that way things could have been clearer. Here the issues are as discussed above, your subproblem is much harder to solve, and what you cited from Nesterov is under the premise (which he actually made but the authors here totally ignore) that the constraint set is a "simple" one. Note that the difference between this work and Accelegrad is that their algorithm takes the set K as something that can be self-defined, and thus in their case the cost of projection is not a problem because one could select a set that is simple, but it is not the case here. And although Accelegrad is for smooth optimization, surely if you select the set K as the constraint set X given in this work, everything will still apply, and thus I wouldn't think the authors' argument against that work is valid. Also, I don't think there's anyway to compare with AcceleGrad fairly: that method deals with unconstrained problems only (as the authors claimed, although this is not true as argued above) while the proposed one deals with constrained problems only.
    I therefore keep my score.*

**Answer:** We respectfully would like to point out that the reviewer’s comments on AcceleGrad are **factually incorrect** for reasons that we will explain below. As a result, we think that there might be some confusion, which we hope to clarify.

AcceleGrad is designed for unconstrained problems and its guarantees are given w.r.t. to an unconstrained sequence. For technical reasons, it strangely requires prior knowledge of some “self-defined” compact set which includes  **the global minimizer** of the objective function; handcrafting such a set implies knowledge of the **global minimizer**. Needless to say that this particular case is directly covered (theoretically) by our current analysis since our choice of $\mathcal{X}$ is generic.

However, the main issue with AcceleGrad handling constrained problems (even simple ones) stems from the choice of their adaptive step-size which accumulates square norms of the gradients:
$$\gamma_{t}=1/\sqrt{\sum_{j=1}^{t}j^{2}\|\nabla f(X_j)\|^{2}} $$

This particular choice aims directly to exploit the fact that the global solution **is** a stationary point, i.e., $\nabla f(x^{*})=0$.

Intuitively, the above step-size suggests that if the method’s iterates approach the solution then the square norm of the gradient tends to zero.
In turn, the growth of the sum of the square norms of the gradients decelerates near the solution. Therefore, the optimizer should eventually afford bigger steps which results in an improved convergence rate.
Similar arguments, of course, apply to its non-accelerated counterpart AdaGrad if run with the simplified Ada-Norm step-size:

$$\gamma_{t}=1/\sqrt{\sum_{j=1}^{t}\|\nabla f(X_j)\|^{2}}$$

That being said, this reasoning immediately collapses for general constrained problems (even for simple ones) for the case where the solution is on the boundary and hence no stationarity is guaranteed .

Now, we strongly believe that from a theoretical point of view the only reasonable comparison of ExtraNewton should be, first and foremost, with their SOTA universal first-order counterparts. Moreover, after the above arguments it should also be clear to the reviewer that considering unconstrained numerical experiments is indispensable if one would like to conduct a fair comparison with AcceleGrad and AdaGrad methods.
However, we are truly open to any suggestions of the reviewer relative to other methodological comparisons.

Therefore, once more we kindly ask the reviewer to reconsider their evaluation. From what we have seen so far, the reviewer’s score is unwarranted as it seems to focus on un-implementability (which we resolved), un-projectability (which we resolved), and unscalability (which is not the focus of the paper as we work on second-order methods). Moreover, we urge the reviewer to keep the conversation within a factual and constructive tone as our paper is a mathematical one and we are not trying to be deceptive (let alone deeply deceptive) at all. After all, this whole process is to advance science, which we hope we do.

---

### Author Response · Authors · 2022-08-08
**A global discussion is requested (2/3)**


- **9QHq:** *I don't see why a stochastic setting here matters: in your experiment you have Newton's method as a baseline for comparison, and what I am saying is simply replacing the operation of computing the exact inverse for solving the linear system (which has nothing to do with whether this is in a stochastic setting or not) with CG. But surely here given a constraint set, (linear) CG is not applicable anymore to the proposed algorithm, so it will get a much higher cost for solving high-dimensional problems over other second-order methods that solve an unconstrained quadratic subproblem. This is also closely related to the 1st point that in many cases, the proposed algorithm is still prohibitively expensive to be useful. If there are restrictions/limitations of the proposed algorithm, the authors should have been clear about that up front.*


**Answer:** We would like to respectfully disagree with this statement of the reviewer for the following reasons.
First, we thank the reviewer for acknowledging that CG has issues with constraints.

Second, could the reviewer also acknowledge that *if* the projections are tractable (also related to your 1st point), the first-order approach to solving subproblems  which we mentioned in our rebuttal (see point 1) is actually an efficient way of solving the problems already? Please note that this is a simple revision where we just clarify that we assume projections are tractable and we contend that this is common practice in constrained optimization problems.

- **9QHq:** *Let me repeat my original question: I also would like to know about the running time comparison of these algorithms, especially in the setting of higher-dimensional problems. The authors' response is still in this low-dimensional dataset that has d=123.*

**Answer:** Perhaps again there is a mismatch in expectations. This work is **not** about solving high-dimensional problems. We develop a new variant of the Newton method that has global rates under acceptable conditions and exhibits rate interpolation guarantees for both deterministic and stochastic oracles; as far as our knowledge goes this is the first of its kind result in the literature.

In the experiments we provided, both the main text and the supplementary material are based on benchmark datasets from LibSVM, which SOTA papers on second-order methods use (see for example, Mischenko (2022), Doikov and Nesterov (2020), Carmon et al., (2022)).

- **9QHq:** *I was only talking about the least-square case in the deterministic setting, namely Fig 1(a). If you solved the subproblem (which is actually the original problem) exactly, any sort of reasonable linesearch procedure will accept unit step size in this case, and surely the initial point doesn't matter, because the solution provides the lowest possible objective. Given the number of data points and the dimensionality of the problem, it is totally not what the authors described. Instead, Given that #instances >> #dimension, it is unlikely the Hessian is only positive semidefinite (which requires all data instances lie in a subspace whose dimension is smaller than the whole problem dimension).*

**Answer:** Could the reviewer  please explain the relevance of their answer to the black-box optimization model, which is what we considered in our original problem formulation? We consider a setting where the algorithm only has access to gradient and Hessian oracles without knowing the structure of the objective.

Moreover, for second-order methods, unlike first-order ones, the line-search procedure starts with a small step-size and then tries to increase it towards 1 (i.e., full step). As a result, we do not start with 1 as the initial step, which the line-search procedure would accept immediately. Having said that, if the reviewer insists, we can simply mention the fact that for the quadratic problem, with step-size 1, the Newton algorithm converges in 1 step. Would that satisfy the reviewer?

---

### Author Response · Authors · 2022-08-08
**A global discussion is requested (1/3)**

We would like to thank all the reviewers for their efforts in reviewing our paper.

We would like to bring everybody on board with the ongoing discussion with Reviewer 9RHq. We appreciate additional feedback from all the reviewers if possible.

We quote the reviewer and provide an answer below.

- **9QHq:** *I thank the authors for their responses to my comments, although I am still not very satisfied with their replies. Here are the major issues that remain for me:
Indeed from the oracle point of view there's no additional cost, but it doesn't mean at all that without an additional oracle call the constrained subproblem becomes easy. In particular, in L. 65, the authors criticized some existing methods for being prohibitively expensive computationally, but this is surely also the case in general for the proposed method. Indeed there are some simple constraint sets such as the box constraints or the norm constraint for which the projection can be conducted easily, and in that situation, the authors' claim could be valid. Unfortunately, the paper boasted that they only need an assumption that the constraint is convex and compact, without any mentioning of the projection/proximal operation. In many situations, even if X is convex and compact, projection onto it could be extremely costly, and thus the proposed method in general is still very expensive, while the authors only suggested special cases in response to my question. And this is also very deceptive that the authors claimed all sort of contributions as if they have really solved the question in L. 38-39, but in fact they can only handle a very small subclass of problems (constrained, with the constraint set compact and convex, and with the constraint set projection easy to compute generally being considered ), while by being deceptive again to conduct experiments in a setting not covered by their theory, this seems to me to mislead people about the contributions of the theoretical part. The authors should make this clear from the beginning, and then the degree of contribution of the new method should surely be reevaluated by all reviewers.*


**Answer:** We observe that the reviewer’s concerns shifted from non-implementability issues (which describe  a completely different problem in the literature) to some general  (and somewhat orthogonal to our theoretical contributions) concerns regarding constrained convex optimization framework.

To that end, we would like to add the following:
Of course, if the optimizer is facing a constrained convex optimization (as we explicitly state at L. 97), the difficulty of solving projections and/or prox-operators  on general convex and compact sets is something well-known throughout the literature of convex optimization and machine learning. To that end, it is a common convention that the geometry of the constraints are projection and/or proximal friendly.

However, this fact does not diminish nor the value neither the contributions of the various  seminal theoretical works ranging from first order methods like:

1. Projected gradient descent (folk, Nemirovski et al (2009) and various follow ups) along with the vast literature concerning their accelerated counterparts ranging from FISTA (Beck and Teboulle (2009)) to various machine learning follow ups like Joulani et al (ICML 2020), Antonakopoulos et al (ICML 2022, oral) and references therein.

2. Mirror Descent and Dual Averaging (where except for some very few exceptions there is no closed update formula), we refer the reviewer to (Beck and Teboulle (2003), Bauschke- Bolte- Teboulle (2017), Nesterov (2009), Shalev -Shwartz (2012))

3. And its extra- gradient counterparts like Extra-Gradient (Korpelevich (1976)), Mirror Prox (Nemirovski (2004)) and Dual Extrapolation (Nesterov (2007))

to second order methods like Cubic Regularization (see Nesterov, Cubic Regularization of Newton's Method for Convex Problems with Constraints, 2006). Note that at this point, as reviewer fZEZ observes (point 4 in their review),  under favorable geometry of $\mathcal{X}$ our method enjoys a similar complexity.

In summary, in order to avoid any misunderstandings, we are more than happy to add a simple statement that projection onto $X$ is efficiently computable, which is indeed what is generally assumed in the literature of constrained optimization as we listed above. We can make this an explicit assumption, which would be easy to incorporate (see our revised version of our paper L 124-125).

---

### Meta-Review · Area_Chair_CmBR · 2022-08-21

**Recommendation:** Accept
**Confidence:** Certain

**Metareview:**

A solid theoretical paper that proposes a second-order method of iteration complexity O(\sigma T^{-1/2} + T^{-3}) for minimizing Hessian-smooth convex functions. A notable feature of the proposed method is its ability to adapt to problem parameters.

The paper has generated considerable discussion between the reviewers and the authors which helped clarifying several major concerns. Please make sure to take into account the important reviewers' feedback, and criticisms, in the revised version.

**Award:**

No

---

### Decision · Program_Chairs · 2022-09-14

Accept